# A Survey on Future Frame Synthesis: Bridging Deterministic and Generative Approaches

**Ruibo Ming**[*]                                                   *mrb22@mails.tsinghua.edu.cn*
*Tsinghua University, StepFun*

**Zhewei Huang**[*]                                                        *hzwer@pku.edu.cn*
*StepFun*

**Jingwei Wu**                                                 *wujingwei22@mails.ucas.ac.cn*
*StepFun*

**Zhuoxuan Ju**                                                       *nymphea@stu.pku.edu.cn*
*Peking University, StepFun*

**Daxin Jiang**                                                          *djiang@stepfun.com*
*StepFun*

**Jianming Hu**                                                     *hujm@mail.tsinghua.edu.cn*
*Tsinghua University*

**Lihui Peng**[†]                                                *lihuipeng@mail.tsinghua.edu.cn*
*Tsinghua University*

**Shuchang Zhou**[†]                                               *shuchang.zhou@gmail.com*
*StepFun, Megvii Technology*

**Reviewed on OpenReview:** *https://openreview.net/forum?id=ZN4rzrHlNz*

## Abstract

Future Frame Synthesis (FFS), the task of generating subsequent video frames from context, represents a core challenge in machine intelligence and a cornerstone for developing predictive world models. This survey provides a comprehensive analysis of the FFS landscape, charting its critical evolution from deterministic algorithms focused on pixel-level accuracy to modern generative paradigms that prioritize semantic coherence and dynamic plausibility. We introduce a novel taxonomy organized by algorithmic stochasticity, which not only categorizes existing methods but also reveals the fundamental drivers—advances in architectures, datasets, and computational scale—behind this paradigm shift. Critically, our analysis identifies a bifurcation in the field's trajectory: one path toward efficient, real-time prediction, and another toward large-scale, generative world simulation. By pinpointing key challenges and proposing concrete research questions for both frontiers, this survey serves as an essential guide for researchers aiming to advance the frontiers of visual dynamic modeling.

## 1 Introduction

The goal of the Future Frame Synthesis (FFS) task is to generate future frames from a sequence of historical frames (Srivastava et al., 2015) or even a single context frame (Xue et al., 2016), optionally guided by supplementary control signals. The learning objective of FFS is also regarded as central to building a world model (Ha & Schmidhuber, 2018a; Hafner et al., 2023). FFS is closely related to low-level computer vision techniques, particularly when synthesizing temporally adjacent frames (Liu et al., 2017; Wu et al., 2022b; Hu et al., 2023b). However, FFS differs from other low-level tasks by implicitly requiring a more sophisticated

---

[*]Both authors contributed equally. [†]Corresponding authors.

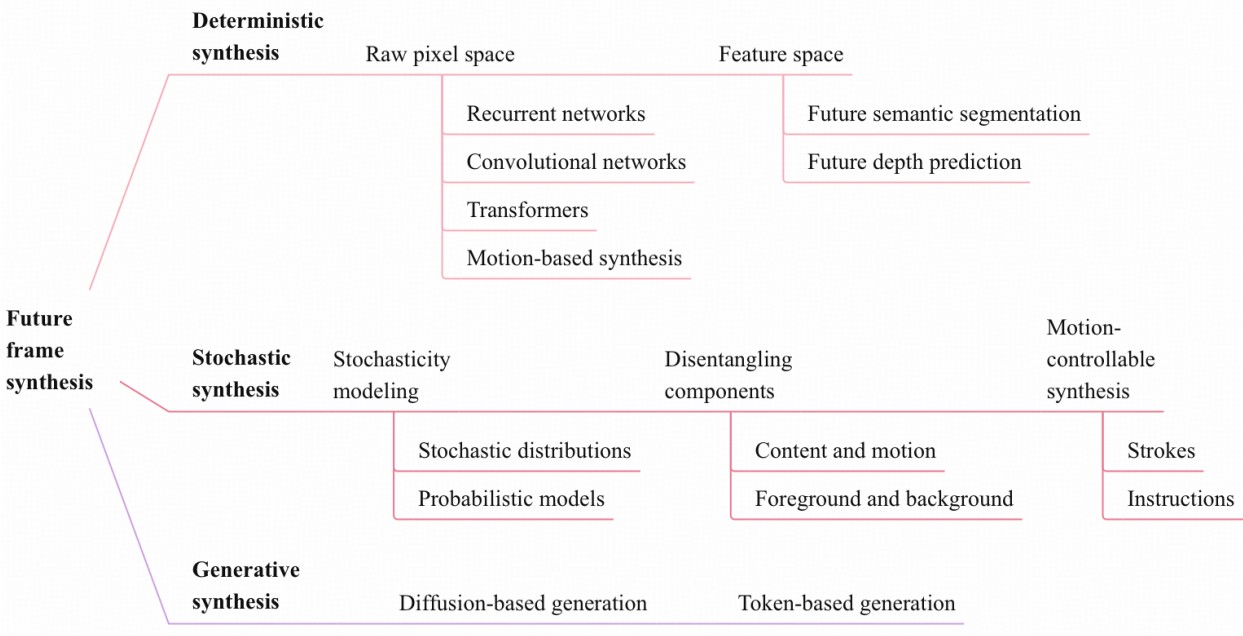

Figure 1: **Structure of our taxonomy.** We categorize future frame synthesis (FFS) approaches based on the degree of stochasticity in the modeling paradigm.

understanding of scene dynamics and temporal coherence—traits typically associated with high-level vision tasks. The key challenge is to design models that can achieve this balance efficiently, using a moderate number of parameters to reduce inference latency and resource consumption, thereby making FFS practical for real-world deployment. This unique positioning underscores the integral role of FFS in bridging the gap between low-level perception & prediction, and high-level understanding & generation in computer vision.

Early FFS methods primarily followed two design approaches, and these algorithms are commonly referred to as video prediction methods. The first approach involves **referencing** pixels from existing frames—typically the last observed frame—to synthesize future content. However, such methods inherently struggle to model the appearance and disappearance of objects. These methods tend to produce accurate short-term predictions but degrade over longer time horizons. The second approach focuses on **generating** future frames from scratch. While these methods hold the potential to model object emergence and disappearance, they primarily operate at the pixel level. As a result, they often fail to capture high-level semantic context, which is essential for realistic and imaginative generation.

Prior to our work, two surveys published in 2020 (Oprea et al., 2020; Rasouli, 2020) provide comprehensive overviews of early technical developments in video prediction. More recently, several surveys have emerged on text-to-video generative models (Liu et al., 2024b) and long video generation (Li et al., 2024; Sun et al., 2024b). In contrast, our survey emphasizes recent advances and explores the interplay between predictive and generative methodologies. We argue that the future of long-term FFS lies in the synergistic integration of prediction and generation techniques. Such a unified approach integrates contextual constraints with semantic understanding, enabling more robust and coherent synthesis.

The field of FFS is rapidly moving from traditional deterministic prediction to large-scale generative approaches as the latest video generative models continue to emerge. These models have made important advances in parameter scales, generation lengths, control capabilities, and training strategies, injecting new vitality. As a foundation, we introduce the problem formulation and core challenges in Section 2. Our taxonomy is organized around the degree of stochasticity in modeling approaches. In Section 3, we present deterministic algorithms that aim to perform pixel-level fitting based on fixed target frames. However, pixel-level metrics tend to drive models to average over multiple plausible futures, often resulting in blurry

predictions. In Section 4, we examine algorithms that enable stochastic motion prediction. These include approaches that inject stochastic variables into deterministic models, as well as methods based on explicit probabilistic modeling. Such methods enable sampling from motion distributions, yielding diverse yet plausible predictions beyond the target frame. Given current FFS algorithms' limited generative capacity, especially for high-resolution videos involving object appearance and disappearance, we discuss the generative FFS methods in Section 5. These methods prioritize producing coherent long-term sequences over pixel-level accuracy. Figure 1 visually illustrates the taxonomy and serves as a structural guide for the subsequent sections.

In Section 6, we explore the broad applicability of FFS in areas such as world model, autonomous driving, robotics, film production, meteorology, and anomaly detection. These use cases demonstrate the role of FFS in dynamic scene understanding and interaction. In Section 7, we review prior surveys on video prediction and diffusion-based video generation. We also clarify our distinct focus: a comprehensive analysis of FFS spanning from deterministic to generative paradigms, emphasizing the growing role of generative models in producing realistic and diverse future predictions.

## 2 Future frame synthesis

### 2.1 Problem definition

The FFS task involves predicting future frames conditioned on previously observed video content. The primary objective is to develop models capable of accurately capturing future visual dynamics. Formally, this task can be formulated as a conditional generative modeling problem: given observed frames $X_{t_1:t_2}$, the goal is to generate future frames $X_{t_2+1:t_3}$. This relationship can be expressed as the conditional probability distribution:

$$X_{t_2+1:t_3} \sim \mathbb{P}(X_{t_2+1:t_3} \mid X_{t_1:t_2}), \tag{1}$$

In Eq. (1), $t_1$ denotes the initial time step, $t_2$ marks the end of the observed frame sequence, and $t_3$ indicates the final time step for FFS. The key challenge is to learn a mapping function that models the complex spatio-temporal dependencies across frames. Here, $\mathbb{P}(X_{t_2+1:t_3} \mid X_{t_1:t_2})$ denotes the conditional probability distribution over future frames given the observed sequence.

Many FFS algorithms incorporate multi-modal data $M_{t_1:t_2}$ including depth maps, landmarks, bounding boxes, and segmentation maps, to enhance scene understanding. They may also include human control signals $C_{t_2+1:t_3}$, such as text instructions or sketch-based strokes, which guide the model to generate future sequences following specific intended trajectories:

$$X_{t_2+1:t_3} \sim \mathbb{P}(X_{t_2+1:t_3} \mid X_{t_1:t_2}, M_{t_1:t_2}, C_{t_2+1:t_3}) \tag{2}$$

In multi-modal prediction methods, such as the recent auto-regressive prediction (Bai et al., 2024; Peng et al., 2024; Ming et al., 2024), multi-modal information might serve as the learning target as well:

$$(X_{t_2+1:t_3}, M_{t_2+1:t_3}) \sim \mathbb{P}(X_{t_2+1:t_3}, M_{t_2+1:t_3} \mid X_{t_1:t_2}, M_{t_1:t_2}, C_{t_2+1:t_3}) \tag{3}$$

### 2.2 Paradigms and architectures

In the domain of FFS, three paradigms have emerged—deterministic, stochastic, and generative—each representing a distinct modeling approach. Early deterministic methods are designed to minimize pixel-wise error. This objective inevitably produces blurry averages, which in turn motivate stochastic formulations that explicitly model uncertainty. Furthermore, the need for long-horizon and high-level plausibility pushes the field toward fully generative paradigms.

The deterministic paradigm emphasizes pixel-level fitting to fixed target frames, typically employing low-level computer vision architectures such as Convolutional Neural Networks (CNNs) (Krizhevsky et al., 2012), Recurrent Neural Networks (RNNs) (Rumelhart et al., 1985), Long Short-Term Memory (LSTM) (Hochreiter & Schmidhuber, 1997) and U-Net (Ronneberger et al., 2015). Recently, the Transformer architecture has gradually challenged the dominance of traditional CNNs and RNNs. A landmark work, Vision Transformer (ViT) (Dosovitskiy et al., 2021), divides images into fixed-size patches and treats these patches as sequences input into the Transformer for processing. Swin Transformer (Liu et al., 2021) further introduces a hierarchical Transformer structure that handles features on different scales through a local window self-attention mechanism and progressively expanding receptive fields. In low-level vision tasks, IPT (Chen et al., 2021) employs pre-trained Transformers to address various low-level visual tasks. And Models like TimeSformer (Bertasius et al., 2021) and ViViT (Arnab et al., 2021) utilize Transformers to process spatio-temporal information in videos by combining video frames and time steps. During this period, models generally pursue efficiency and smaller parameter scales. This is constrained by the prevailing GPU computing power at the time and aligns with their primary goal of addressing short-term, low-resolution prediction tasks.

For deterministic models, optimizing pixel-level metrics such as Peak Signal-to-Noise Ratio (PSNR) and Structural Similarity (SSIM) (Wang et al., 2004) often leads to blurry outputs, as the models tend to average over multiple plausible futures. Regarding the evaluation metrics, we will delve into the discussion in the next subsection. Unlike deterministic models that usually optimize pixel-level metrics, the stochastic paradigm introduces randomness into the prediction process by incorporating stochastic variables or distributions to model the inherent uncertainty in video dynamics. This approach aims to capture the variability and unpredictability of video sequences, often producing results that deviate significantly from the ground truth. Probabilistic models, such as Variational Autoencoders (VAEs) (Kingma & Welling, 2014) and Generative Adversarial Networks (GANs) (Goodfellow et al., 2020), are commonly used to achieve this. This approach does not necessarily aim to generate entirely new video content but rather to capture the variability in future outcomes. The generative paradigm, on the other hand, prioritizes the synthesis of coherent and plausible video sequences over pixel-level fidelity. It leverages advanced generative models, such as diffusion models and large language models (LLMs), to produce diverse and imaginative future frames that capture complex scene dynamics, including object emergence and disappearance. Recently, diffusion models have emerged as a powerful paradigm for generative tasks, including FFS. These models, such as DDPM (Ho et al., 2020), learn to generate data by progressively denoising a random noise signal. Diffusion models like Video Diffusion (Ho et al., 2022) and SVD (Blattmann et al., 2023a) have demonstrated remarkable ability to produce high-quality, diverse, and temporally coherent video sequences. By leveraging the gradual denoising process, these models can effectively capture complex scene dynamics and generate realistic future frames, even for long-term predictions. Furthermore, flow-matching models (Lipman et al., 2022; Dao et al., 2023) and rectified flow (Liu et al., 2022) have gained attention as an alternative approach to generative modeling, offering improved efficiency and scalability compared to traditional diffusion models. The rise of diffusion models and large auto-regressive models is inextricably linked to the significant growth in computational resources in recent years and the widespread utilization of large-scale, high-resolution web video datasets. As research evolves, the boundaries between these paradigms continue to blur, with growing efforts to integrate their strengths and build more capable and versatile FFS systems.

## 2.3 Overall challenges and development trends

The field of FFS faces several longstanding challenges, including the need for algorithms that balance low-level pixel fidelity with high-level scene understanding, the lack of reliable perceptual and stochastic evaluation metrics, the difficulty of achieving long-term synthesis, and the scarcity of high-quality high-resolution datasets that capture stochastic motion and object emergence and disappearance. This section outlines these key challenges and sets the stage for further discussion.

### 2.3.1 Learning objectives and evaluation metrics

Low-level metrics such as PSNR and SSIM assess only the pixel-wise accuracy of predictions. To optimize for these metrics, models are typically trained using pixel-space losses such as $\ell_1$ or $\ell_2$. They often lead to blurry predictions that closely match the ground truth, rather than sharper and more realistic generations

that deviate from it—a phenomenon known as the perception-distortion trade-off (Blau & Michaeli, 2018). As a result, researchers are increasingly exploring alternative evaluation metrics, including perceptual metrics (e.g., DeePSiM (Dosovitskiy & Brox, 2016), LPIPS (Zhang et al., 2018)) and stochastic metrics (e.g., IS (Salimans et al., 2016), FID (Heusel et al., 2017)). These metrics are believed to better align with human perceptual judgments. However, even classifiers trained on human-annotated perceptual data show limited agreement with human judgments of image quality (Kumar et al., 2022).

In visual domains, models are typically evaluated based on the perceptual quality of their generated outputs. However, for many practical applications, perceptual quality may not be the most critical factor. For instance, Dreamer-V3 (Hafner et al., 2023) and VPT (Baker et al., 2022) have successfully built effective world models using low-resolution frame sequences. Moreover, most visual representation learning methods are developed using relatively low-resolution images (Radford et al., 2021; He et al., 2022). We are concerned that an excessive pursuit of visual quality may bias model selection toward architectures that overfit low-level features. Beyond aligning with human perceptual judgments, evaluation metrics should also be designed to assess a model's capacity to capture scene dynamics and temporal variations. Moreover, we also need to focus on how to leverage the capabilities of the FFS model to assist us in accomplishing more tasks (Agarwal et al., 2025).

Regarding the quality assessment of video generative models, some more comprehensive evaluation systems have been recently established. VBench (Huang et al., 2024) benchmark suite has been proposed to address these concerns by providing a comprehensive evaluation framework for video generative models. VBench evaluates video generative models comprehensively, covering perceptual quality, dynamics, temporal consistency, content diversity, and prompt alignment for a holistic performance assessment. FVMD (Liu et al., 2024a) designs explicit motion features based on key point tracking, focusing on evaluating motion consistency in video generation. VBench-2.0 (Zheng et al., 2025) further focuses on the intrinsic faithfulness of generative models, such as whether they adhere to real-world principles, including physical laws and common sense reasoning. Recently, some of the work has increasingly utilized multi-modal LLMs to evaluate the generated results. For example, ViStoryBench (Zhuang et al., 2025) leverages GPT-4o (Hurst et al., 2024) to assess the instruction-following ability generative models from multiple aspects. As people's pursuit of generating realism gradually increases, the evaluation of real physical interaction and real-scene simulation will be a promising research topic. Moreover, for FFS, we need to consider whether the generated results are reasonable given the observed results.

Even with improved evaluation metrics, optimizing them during training remains a significant challenge. During training, researchers often use pre-trained ImageNet classifiers as feature extractors (Johnson et al., 2016; Kumar et al., 2022) to compare generated outputs with ground truth, thereby optimizing for both low-level and high-level features. Additionally, various GAN-based loss functions have been proposed to enhance the perceptual quality of generated outputs (Huang et al., 2017; Zhang et al., 2020b). Diffusion models seem to bring significantly stronger realism, but new issues such as slow convergence and high resource consumption still require people to invest effort in exploring.

### 2.3.2   Long-term synthesis

Despite significant progress in short-term video prediction, synthesizing events over extended time horizons remains challenging due to long-term dependencies and complex object interactions in dynamic scenes. Naive application of short-term models in an iterative manner often leads to rapid quality degradation (Wu et al., 2022b; Hu et al., 2023b). Low-level vision methods such as DMVFN (Hu et al., 2023b) might primarily focus on video prediction for a limited number of future frames, typically within 0.3 seconds. Owing to limited model capacity for forming a comprehensive understanding of the real world, most existing video synthesis models primarily model pixel-level distributions. When dealing with natural videos over long durations, these models struggle to predict object dynamics while preserving visual quality. One promising direction is to incorporate high-level structural information (Villegas et al., 2017; Ming et al., 2024). Leveraging such higher-order representations helps models retain key details and maintain temporal consistency over extended time scales. Utilizing the prior of large diffusion models, such as SVD (Blattmann et al., 2023a), enables the generation of high-definition videos ranging from 2 to 4 seconds. However, we need appropriate techniques to preserve the capabilities of these pre-trained models while unlocking their conditional gen-

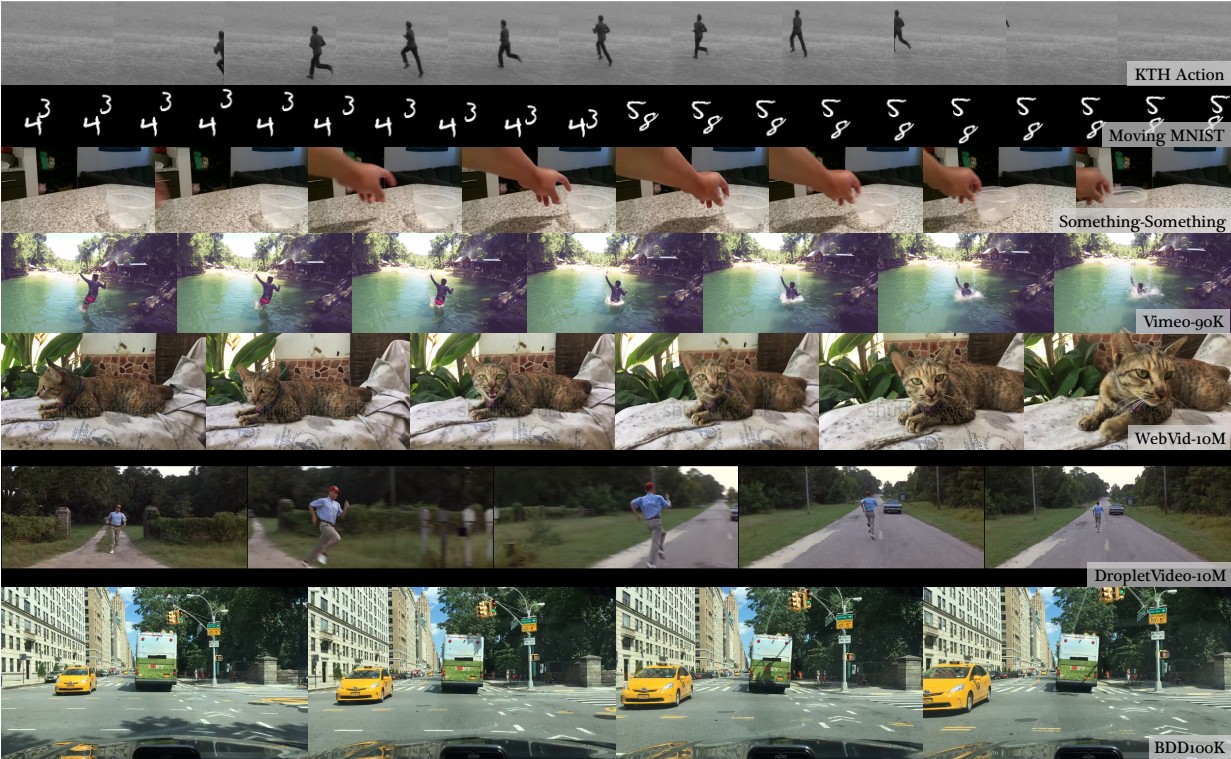

Figure 2: **Dataset samples.** We adjust the absolute resolution of the image for visualization while maintaining the relative size relationships.

eration functionalities (Wang et al., 2025b; Yang et al., 2025). When generating longer videos, properly handling physical interactions presents greater challenges (Agarwal et al., 2025).

### 2.3.3 Generalization

The interplay between data volume and model complexity jointly determines the upper bounds of an algorithm's performance. Despite the vastness of video data available on the Internet, the scarcity of high-quality video datasets suitable for video synthesis remains a limiting factor. Existing datasets often suffer from simplistic data distribution, low resolution, and limited motion diversity. These limitations hinder the ability of current video synthesis models to handle high-resolution content and large motion scales, thus restricting their practical utility to diverse and unseen scenarios. High-resolution video synthesis is inherently challenging and requires substantial computational resources (Blattmann et al., 2023b). The computational burden further complicates the real-time deployment in practical applications.

### 2.4 Datasets

The advancement of video synthesis models is highly dependent on the diversity, quality, and characteristics of training datasets. A common observation is that the suitability of datasets varies with their dimensionality and size: lower-dimensional datasets, often smaller in scale, tend to exhibit limited generalization. Conversely, higher-dimensional datasets offer greater variability, contributing to stronger generalization capabilities. Generally, video complexity and resolution have increased in recent years, as shown in Figure 2. In Table 1, we summarize the most widely used datasets in video synthesis, highlighting their scale and available supervisory signals to provide a comprehensive overview of the current dataset landscape. For datasets lacking detailed reports in their original papers or project pages, we estimate missing values using mean or median statistics to ensure consistency across the analysis.

| Dataset | Category | # Videos | # Clip Frames | Resolution | Extra Annotations |
|---|---|---|---|---|---|
| KTH Action (Schuldt et al., 2004) | Human | 2, 391 | 95* | 160 × 120 | Class |
| Caltech Pedestrian (Dollar et al., 2011) | Human | 137 | 1, 824 | 640 × 480 | Bounding Box |
| HMDB51 (Kuehne et al., 2011) | Human | 6, 766 | 93* | 414 × 404† | Class |
| SJTU 4K (Song et al., 2013) | General | 15 | 300 | 3840 × 2160 | - |
| KITTI (Geiger et al., 2013) | Traffic | 151 | 323* | 1242 × 375 | OF, BBox, Sem, Ins, Depth |
| J-HMDB (Jhuang et al., 2013) | Human | 928 | 34* | 320 × 240 | OF, Ins, HJ, Class |
| Penn Action (Zhang et al., 2013) | Human | 2, 326 | 70* | 480 × 270† | Human Joint, Class |
| Moving MNIST (Srivastava et al., 2015) | Simulation | 10, 000 | 20 | 64 × 64 | - |
| MSR-VTT (Xu et al., 2016) | General | 10, 000 | 408* | 320 × 240 | Class, Text |
| Cityscapes (Cordts et al., 2016) | Traffic | 46 | 869* | 2048 × 1024 | Semantic, Instance, Depth |
| DAVIS17 (Pont-Tuset et al., 2017) | General | 150 | 73* | 3840 × 2026† | Semantic |
| SM-MNIST (Denton & Fergus, 2018) | Simulation | customize | customize | 64 × 64 | - |
| nuScenes (Caesar et al., 2020) | Traffic | 1, 000 | 40* | 1600 × 900 | BBox, Semantic |
| QST (Zhang et al., 2020a) | Sky | 1, 167 | 245* | 1024 × 1024 | - |
| X4K1000FPS (Sim et al., 2021) | General | 4, 408 | 65* | 4096 × 2160 | - |
| Youtube Driving (Zhang et al., 2022) | Traffic | 134 | variable | variable | Action |
| OpenDV-2K (Yang et al., 2024) | Traffic | 2, 139 | variable | variable | Text |
| ChronoMagic (Yuan et al., 2025b) | General | 2, 265 | variable | 1920 × 1080 | Text |
| UCF101 (Soomro et al., 2012) | Human | 13, 320 | 187* | 320 × 240 | Class |
| Sports-1M (Karpathy et al., 2014) | Human | 1, 133, 158 | variable | variable | Class |
| Robotic Pushing (Finn et al., 2016) | Robot | 59, 000 | 25* | 640 × 512 | Class |
| YouTube-8M (Abu-El-Haija et al., 2016) | General | 8, 200, 000 | variable | variable | Class |
| Something-Something (Goyal et al., 2017) | Object | 220, 847 | 45 | 427 × 240† | Text |
| DiDeMo (Anne Hendricks et al., 2017) | Human | 10, 464 | variable | variable | Text |
| Sky Time-lapse (Xiong et al., 2018) | Sky | 38, 207 | 32 | 640 × 360 | - |
| ShapeStacks (Groth et al., 2018) | Simulation | 36, 000 | 16 | 224 × 224 | Semantic |
| Vimeo-90K (Xue et al., 2019) | General | 91, 701 | 7 | 448 × 256 | - |
| D²-City (Che et al., 2019) | Traffic | 11, 211 | 750* | 1080p / 720p | BBox |
| HowTo100M (Miech et al., 2019) | General | 136, 600, 000 | variable | variable | Text |
| Kinetics-700 (Carreira et al., 2019) | Human | 650, 000 | 250* | variable | Class |
| RoboNet (Dasari et al., 2019) | Robot | 161, 000 | 93* | 64 × 48 | - |
| BDD100K (Yu et al., 2020) | Traffic | 100, 000 | 1175* | 1280 × 720 | BBox, Semantic, Depth |
| VGG-Sound (Chen et al., 2020) | General | 199, 467 | variable | variable | Class, Audio |
| WebVid-10M (Bain et al., 2021) | General | 10, 732, 607 | 449* | 596 × 336 | Text |
| Time-lapse-D (Xue et al., 2021) | Sky | 16, 874 | variable | variable | Class |
| HD-VILA-100M (Xue et al., 2022) | General | 100, 000, 000 | variable | 1280 × 720 | Class, Text |
| CelebV-HQ (Zhu et al., 2022) | Facial | 35, 666 | variable | 512 × 512 | Action |
| CelebV-Text (Yu et al., 2023a) | Facial | 70, 000 | variable | 512 × 512 | Action, Text |
| SportsSlomo (Chen & Jiang, 2024) | Human | 130, 000 | 7 | 1280 × 720 | - |
| Pandas-70M (Chen et al., 2024c) | General | 70, 000, 000 | variable | 1280 × 720 | Class, Text |
| InternVideo2 (Wang et al., 2024d) | General | 2, 000, 000 | variable | variable | Action, Text |
| VidGen-1M (Tan et al., 2024) | General | 1, 000, 000 | variable | 1280 × 720 | Text |
| ConsisID (Yuan et al., 2025a) | Human | 54, 239 | 129* | 1280 × 720 | Text |
| DropletVideo-10M (Zhang et al., 2025) | General | 10, 000, 000 | variable | variable | Text |

*  denotes the mean value. † denotes the median value. **OF**: Optical Flow, **BBox**: Bounding Box, **Sem**: Semantic, **Ins**: Instance, **HJ**: Human Joints)

Table 1: Summary of commonly used FFS datasets, including dataset category, total number of videos, frame count for each video clip, image resolution, and additional annotations.

**Challenges.** 1. Unifying the organization of image and video data. A large proportion of computer vision research has historically focused on the image modality. As a result, image datasets are often more carefully curated and contain richer annotations. Representative large-scale image datasets include YFCC100M (Thomee et al., 2016), WIT400M (Radford et al., 2021), and LAION400M (Schuhmann et al., 2021). Given the scale of available image data, it is important to effectively leverage knowledge from foundational image models. When incorporating video data into training pipelines, it is often necessary to filter out low-quality segments and select an appropriate sampling frame rate.

2. Determining the proportion of data from different domains. Computer graphics composite data, 2D anime data, real-world videos, and videos with special effects exhibit vastly different visual characteristics. Moreover, standardizing data from diverse sources to a fixed resolution is challenging due to varying aspect ratios and the presence of resolution-dependent details such as subtitles and textures. Many frame synthesis

methods are sensitive to resolution partly due to the correlation between resolution and object motion intensity (Sim et al., 2021; Hu et al., 2023b; Yoon et al., 2024).

## 3 Deterministic synthesis

| Method | Publication | Main Ideas |
|---|---|---|
| ConvLSTM (Shi et al., 2015) | NeurIPS'15 | Formulate precipitation nowcasting as a spatio-temporal sequence forecasting problem, propose the convolutional LSTM to build an end-to-end model. |
| PredNet (Lotter et al., 2017) | ICLR'17 | Use a recurrent CNN with both bottom-up and top-down connections, with each neural layer making local residual predictions. |
| PredRNN (Wang et al., 2017) | NeurIPS'17 | Introduce a new spatio-temporal LSTM unit, that extracts and memorizes spatial and temporal representations simultaneously. |
| E3d-LSTM (Wang et al., 2019) | ICLR'19 | Integrate 3D convolutions into RNNs to enhance local motion modeling, and make the present memory state interact with its long-term historical records. |
| MSPred (Villar-Corrales et al., 2022) | BMVC'22 | Focus on enhancing long-term action planning ability, and use spatio-temporal downsampling to forecast at different scales. |
| Multi-Scale AdvGDL (Mathieu et al., 2016) | ICLR'16 | Propose three feature learning strategies to address blurriness in FFS: a multi-scale architecture, adversarial training, and an image gradient difference loss. |
| PredCNN (Xu et al., 2018) | IJCAI'18 | Design a cascade multiplicative unit that provides more operations for previous frames, and capture the temporal dependencies through stacked operations. |
| DVF (Liu et al., 2017) | ICCV'17 | Learn to synthesize video frames by warping pixel values from existing ones, and unify video interpolation and extrapolation within a single CNN framework. |
| SimVP (Gao et al., 2022) | CVPR'22 | Propose a simple CNN video prediction model without complicated tricks, and provide insights for selecting different architectures. |
| SDC-Net (Reda et al., 2018) | ECCV'18 | Learn a motion vector and a kernel for each pixel to synthesize the predicted pixel, and inherit the merits of both vector-based and kernel-based approaches. |
| FVS (Wu et al., 2020) | CVPR'20 | Decouple the background scene and moving objects, construct future frames using non-rigid deformation of the background and affine transformation of objects. |
| OPT (Wu et al., 2022b) | CVPR'22 | Solve FFS as an optimization problem, with a pre-trained VFI module to construct function. By eliminating the domain gap, OPT is robust in general scenarios. |
| DMVFN (Hu et al., 2023b) | CVPR'23 | On the basis of DVF, multi-scale coarse-to-fine prediction is added, and the input is processed by a dynamic routing subnetwork at the inference stage. |
| VPTR Ye & Bilodeau (2023) | IVC'23 | Present a new efficient building block of Transformer-based models for FFS, along with three competitive variants for video prediction. |
| S2S (Luc et al., 2017) | ICCV'17 | Introduce the task of predicting semantic segmentations of future frames, and show that predicting future segmentations is substantially better than segmenting predicted frames. |
| SADM (Bei et al., 2021) | CVPR'21 | Decompose the scene layout (semantic map) and motion (optical flow) into layers, and hope to explicitly represent objects and learn their class-specific motion. |
| MAL (Liu et al., 2023) | WACV'23 | Predict the depth maps of future frames using a two-branch structure. One branch handles future depth estimation, and the other aids image reconstruction. |

Table 2: Overview of deterministic synthesis methods.

### 3.1 Raw pixel space

In short-term FFS, approaches operating in the raw pixel space have achieved promising results. In this section, we review representative methods and discuss the associated challenges. We will first introduce the evolution of the architecture of popular backbones, and then highlight one of the most popular methods in video prediction in pixel space, motion-based synthesis.

#### 3.1.1 Recurrent networks

Due to the inherent temporal nature of video data, RNNs became a natural choice for early future frame synthesis tasks. The core idea of such methods is to use RNN's memory units to model the temporal dependencies between frames. However, the main challenge lies in how to effectively extend RNN's ability to process one-dimensional sequences to high-dimensional spatio-temporal data like videos. PredNet (Lotter et al., 2017) pioneered the exploration of recurrent neural networks in video synthesis, drawing inspiration from predictive coding in neuroscience and employing a recurrent convolutional network to process video features effectively. Building on this foundation, PredRNN (Wang et al., 2017) introduces significant improvements by modifying LSTM architecture with a dual-memory structure, aiming to enhance spatio-temporal modeling. Despite these advances, the model still faces challenges such as gradient vanishing in video synthesis tasks.

To address these limitations, ConvLSTM (Shi et al., 2015) emerges as a pivotal model by ingeniously integrating LSTM with CNNs to effectively capture motion and spatio-temporal dynamics—a development that has significantly influenced subsequent video synthesis models. E3d-LSTM (Wang et al., 2019) further advances the field by incorporating 3D convolutions into RNNs and introducing a gate-controlled self-attention module, thereby significantly improving long-term synthesis capabilities. However, the increased computational complexity introduced by 3D convolutions may offset the performance gains in certain applications. MSPred (Villar-Corrales et al., 2022) proposes a hierarchical convolutional-recurrent network that operates at multiple temporal frequencies to predict future video frames, as well as other representations such as poses and semantics.

**Challenges.** Despite their effectiveness in capturing temporal dependencies, recurrent networks face several challenges in video prediction tasks. Their inherently sequential nature, which enables frame-by-frame modeling, can result in high computational complexity—particularly in high-resolution scenarios. This is evident from the significantly higher FLOPs and lower FPS observed in recurrent-based models compared to their recurrent-free counterparts (Tan et al., 2023). Additionally, recurrent networks are prone to gradient vanishing and exploding problems, which can severely hinder their ability to learn long-term dependencies (Gao et al., 2022). These challenges underscore the need for alternative approaches that balance efficiency and performance—such as recurrent-free models, which have demonstrated promising results across various video prediction tasks.

### 3.1.2 Convolutional networks

CNNs play an instrumental role in the evolution of video synthesis technology. The progress began with Multi-Scale AdvGDL (Mathieu et al., 2016), initiating a series of significant advances in the field. Following this, PredCNN (Xu et al., 2018) establishes a new benchmark by outperforming its predecessor PredRNN (Wang et al., 2017) across various datasets. The introduction of SimVP (Gao et al., 2022) marks another milestone in convolutional approaches to video prediction. Inspired by the advances of ViT (Dosovitskiy et al., 2021), this approach introduces a simplified CNN architecture to extract continuous tokens, demonstrating that such a configuration can achieve comparable performance in video synthesis.

**Challenges.** Although simple to implement and fast, pure 2D CNN-based models are not well-suited for spatially shifting the pixels of input frames. CAIN (Choi et al., 2020) and FLAVR (Kalluri et al., 2023) respectively introduce channel attention and 3D U-Net architectures for intermediate frame synthesis, but they do not fully replace explicit pixel motion approaches such as kernel-based and flow-based methods. Moreover, in pursuit of efficiency, most CNN-based models used for FFS maintain a relatively small parameter count, typically under 60M (Tan et al., 2023). By contrast, video diffusion models have been scaled up to over 1.5B parameters (Blattmann et al., 2023a) in order to fully leverage large-scale datasets. Performing effective scaling up CNN-based models remains a significant challenge. It is speculated that short-term prediction models for high-resolution, real-time applications and those aiming to leverage large datasets for enhanced generation capabilities will follow diverging development paths.

### 3.1.3 Transformers

Since the groundbreaking design of ViT (Dosovitskiy et al., 2021), which applied a pure Transformer directly to sequences of image patches to extract continuous tokens, the use of Transformers in frame synthesis has attracted significant attention. Video frame interpolation is a task closely related to FFS (Liu et al., 2017). Shi et al. (2022) and Lu et al. (2022) propose Transformer-based video interpolation frameworks that leverage self-attention mechanisms to capture long-range dependencies and enhance content awareness. The primary difference from convolution is that the self-attention mechanism can dynamically compute the correlation between each patch and all other patches, thereby capturing global information, whereas convolution extracts features within local regions using fixed kernels. These methods further introduce innovative strategies, such as local attention in the spatio-temporal domain and cross-scale window-based attention, to improve performance and effectively handle large motions. Ye & Bilodeau (2023) present an efficient Transformer model for video prediction, leveraging a novel local spatio-temporal separation attention mechanism, and compares three variants: fully auto-regressive, partially auto-regressive, and non-auto-regressive, to balance

performance and complexity. Numerous ongoing studies continue to explore the use of Transformers for modeling inter-frame dynamics under varying motion amplitudes and for addressing challenges in high-resolution frame synthesis (Park et al., 2023; Zhang et al., 2024).

**Challenges.** Many researchers believe that Transformers perform particularly well on large-scale datasets (Zhai et al., 2022; Smith et al., 2023). However, most of the existing studies focus primarily on high-level vision tasks. Furthermore, the success of Transformers in many tasks depends heavily on fully leveraging the capabilities of foundation models. In the context of image synthesis, best practices remain unclear (Li et al., 2023). These challenges also explain why generative methods, especially token-based methods that directly draw on LLM architectures and training paradigms, which we will explore in Section 5.2, have become a more attractive direction for exploration.

### 3.1.4 Motion-based synthesis

Optical flow is a technique used to describe the movement of pixels between consecutive frames in a video. By calculating the displacement of each pixel from one frame to the next, optical flow enables the warping of pixels from the current frame to generate synthetic frames that represent the near future (Liu et al., 2017). Flow-based methods can be viewed as complements of kernel-based approaches (Niklaus et al., 2017), since the latter typically constrain the motion of pixel to a relatively small neighborhood (Cheng & Chen, 2021). SDC-Net (Reda et al., 2018) proposes a hybrid approach that inherits the strengths of both vector-based and kernel-based methods. FVS (Wu et al., 2020) enhances synthesis quality by incorporating supplementary information, such as semantic maps and instance maps from input frame sequences. While effective, this method introduces challenges due to increased data modalities and higher computational demands. OPT (Wu et al., 2022b) estimates optical flow through an optimization-based approach. By iteratively refining the current optical flow estimation, the quality of the next frame can be significantly improved. This approach effectively leverages knowledge from the off-the-shelf optical flow models (Teed & Deng, 2020) and video frame interpolation methods (Jiang et al., 2018; Huang et al., 2022b). Although training is not required, the iterative optimization process during inference incurs substantial computational cost. DMVFN (Hu et al., 2023b) improves dense voxel flow (Liu et al., 2017) estimation by dynamically adapting the network architecture based on motion magnitude. DMVFN further confirms the effectiveness of a coarse-to-fine, multi-scale, end-to-end model in addressing short-term motion estimation.

**Challenges.** Optical flow estimation remains an active and extensively studied research topic (Teed & Deng, 2020; Huang et al., 2022a; Sun et al., 2022; Dong & Fu, 2024). However, mainstream optical flow models trained on synthetic data with heavy augmentations often diverge significantly from the scenarios targeted by FFS. Moreover, the learning objectives of these models are not aligned with the goal of generating optical flow that facilitates accurate pixel warping and high-quality image synthesis. Xue et al. (2019) point out that for different downstream tasks, it is often necessary to fine-tune or even train a flow estimation network from scratch. Interestingly, higher-performing optical flow networks may lead to degraded image synthesis results, as they often emphasize ambiguous regions such as occlusions and may lack sufficient spatial resolution (Niklaus & Liu, 2020; Huang et al., 2022b). In real-world scenarios, obtaining ground-truth optical flow labels remains a major challenge.

We believe that near-future frames can be synthesized with increasing accuracy as optical flow methods continue to improve. However, integrating optical flow methods into long-term video generation remains a significant challenge (Liang et al., 2024). Optical flow is typically limited to predicting pixel motion over very short time spans and cannot assist in generating novel video content.

There are still ample opportunities for exploration in the representation of motion. For example, GaussianPrediction Zhao et al. (2024) explores the use of 3D Gaussian representations to model the appearance and geometry of dynamic scenes, enabling image rendering for future scenarios. MoSca (Lei et al., 2024) proposes the Motion Scaffold representation, which lifts the predictions of 2D foundational models (such as depth estimation, pixel trajectories, and optical flow) to 3D and optimizes them using physics-inspired constraints. Ultimately, MoSca reconstructs the geometry and appearance of the scene with a set of dynamic Gaussians (Wu et al., 2024b). Building on video diffusion models, V3D (Chen et al., 2024g) leverages their

learned world simulation capabilities to perceive the 3D world, introduces a geometrical consistency prior, and finetunes the video diffusion model into a multi-view consistent 3D generator. Seamlessly integrating new representation methods with existing frameworks presents novel challenges.

### 3.2 Feature space

Synthesizing in the raw pixel space often overburdens models, as it requires reconstructing images from scratch—a task especially challenging for high-resolution video datasets. This challenge has prompted a shift in focus among researchers. Rather than grappling with the complexities of pixel-level synthesis, many studies have shifted toward high-level feature synthesis in the feature space, focusing on representations such as segmentation maps and depth maps. These approaches provide a more efficient means of handling the complexities of video data (Oprea et al., 2020).

**Future semantic segmentation.** Future semantic segmentation represents a progressive approach to video synthesis, primarily focusing on generating semantic maps for future video frames. This methodology departs from traditional raw pixel forecasting by utilizing semantic maps to narrow the synthesis scope and enhance scene understanding. In this context, the S2S model (Luc et al., 2017) emerges as a pioneering end-to-end system. It processes RGB frames along with their corresponding semantic maps as both input and output. This integration not only advances future semantic segmentation but also enhances video frame prediction, demonstrating the advantages of semantic-level forecasting. Building on this foundation, SADM (Bei et al., 2021) introduces further innovation by integrating optical flow with semantic maps. This fusion leverages optical flow for motion tracking and semantic maps for appearance refinement, using the former to warp input frames and the latter to inpaint occluded regions.

**Future depth prediction.** Depth maps, as 2D data structures that encode 3D information, can provide models with enhanced perception of the 3D world at minimal computational cost. Predicting future depth maps can benefit FFS tasks. MAL (Liu et al., 2023) introduces a meta-learning framework with a two-branch architecture, comprising future depth prediction and an auxiliary image reconstruction task. This framework improves the quality of synthesized future frames, particularly in complex and dynamic scenes.

**Challenges.** Future prediction in the feature space presents significant challenges due to the complex interplay between temporal dynamics and spatial context. Models must capture intricate motion patterns and accurately predict depths or semantic regions, which requires a deep understanding of 3D scene structures and object interactions. Ensuring temporal consistency and spatial precision while handling occlusions, perspective changes, and complex backgrounds is crucial. The use of high-resolution feature maps and large-scale annotated datasets further increases computational and data demands. Generalizing to unseen scenes and objects remains a major challenge, requiring robust models capable of adapting to diverse visual appearances and contexts. These challenges underscore the need for innovative approaches, such as meta-auxiliary learning, to enhance future prediction capabilities.

## 4 Stochastic synthesis

In its early stages, video synthesis was primarily regarded as a low-level computer vision task, with an emphasis on using deterministic algorithms to optimize pixel-level metrics such as MSE, PSNR, and SSIM. However, this approach inherently limits the creative potential of such models by constraining possible motion outcomes to a single, fixed trajectory (Oprea et al., 2020). In response to this limitation, the field of video synthesis has undergone a paradigm shift—from relying on short-term deterministic prediction to embracing long-term stochastic generation. This transition acknowledges that, although stochastic synthesis may produce results that deviate significantly from the ground truth, it is essential for fostering a deeper understanding and enhancing creativity in the modeling of video evolution.

In this section, we explore stochastic synthesis methods, including GANs and VAEs, which were originally categorized for their emphasis on modeling randomness and uncertainty in video motion. Although these models possess generative capabilities, they were originally more closely associated with stochasticity due to

| Method | Publication | Main Ideas |
|---|---|---|
| VPN (Kalchbrenner et al., 2017) | ICML'17 | Model the temporal, spatial, and color structure of video tensors, and encode it as a four-dimensional dependency chain. |
| SV2P (Babaeizadeh et al., 2018) | ICLR'18 | Propose a stochastic variational video prediction (SV2P) method, providing effective stochastic multi-frame prediction for real-world videos. |
| PFP (Hu et al., 2020) | ECCV'20 | Introduce a conditional variational approach to model the stochasticity. Learn a representation that can be decoded to future segmentation, depth and optical flow. |
| SRVP (Franceschi et al., 2020) | ICML'20 | Introduce a stochastic temporal model using a residual update rule in a latent space, inspired by differential equation discretization schemes. |
| PhyDNet (Guen & Thome, 2020) | CVPR'20 | Disentangle PDE dynamics from unknown complementary information, and propose a recurrent physical cell to perform PDE-constrained prediction in latent space. |
| TPK (Walker et al., 2017) | ICCV'17 | Decompose video prediction: Use a VAE to model future human poses, and a GAN to generate future frames conditioned on these poses. |
| SVG (Denton & Fergus, 2018) | ICML'18 | Learn a prior model of uncertainty to generate frames by drawing samples, and combine them with a deterministic estimate to generate varied and sharp results. |
| Vid2Vid (Wang et al., 2018) | NeurIPS'18 | Model a mapping function from a source video (e.g., segmentation) to a photorealistic video, using carefully-designed GAN and a spatio-temporal adversarial objective. |
| SAVP (Lee et al., 2018) | preprint | Combine the latent variational variable model and adversarially-trained models, to produce predictions that look more realistic and better cover the range of possible futures. |
| Retrospective Cycle GAN (Kwon & Park, 2019) | CVPR'19 | Employ two discriminators to identify fake frames and fake contained image sequences. Predict future and past frames while enforcing the consistency of bi-directional prediction. |
| vRNN (Castrejon et al., 2019) | ICCV'19 | Argue that the blurry predictions of VAEs may caused by underfitting, and suggest increasing expressiveness of the latent distributions and using higher capacity likelihood models. |
| GHVAE (Wu et al., 2021) | CVPR'21 | Attempt to address the underfitting issue on large and diverse datasets, by greedily training each level of a hierarchical autoencoder to learn high-quality video predictions. |
| INR-V (Sen et al., 2022) | TMLR'22 | Propose a video representation network that utilizes INRs and a meta-network to generate diverse novel videos, demonstrating superior performance in various video-based generative tasks |
| DIGAN (Yu et al., 2022) | ICLR'22 | Introduce INRs into GAN to enhance motion dynamics, and utilize a motion discriminator that effectively detects unnatural motions. |
| StyleGAN-V (Skorokhodov et al., 2022) | CVPR'22 | Extend the paradigm of neural representations to build a continuous-time generator, and design a holistic discriminator that aggregates temporal information by concatenating frames' features. |
| CDNA (Finn et al., 2016) | NeurIPS'16 | Develop an action-conditioned video prediction model that explicitly models pixel motion. The model generalizes to unseen objects by decoupling motion and appearance. |
| AMC-GAN (Jang et al., 2018) | ICML'18 | By providing appearance and motion information as conditions, reduce the prediction uncertainty of equally probable future. |
| MoCoGAN (Tulyakov et al., 2018) | CVPR'18 | Generate a video by mapping a sequence of random vectors to a sequence of video frames, and introduce a novel adversarial learning scheme utilizing both image and video discriminators. |
| LMC (Lee et al., 2021) | CVPR'21 | Study how to store abundant long-term contexts, and recall suitable motion context, especially complex motions from limited inputs. |
| SLAMP (Akan et al., 2021) | ICCV'21 | Focus on learning stochastic variables for separate content and motion. |
| MMVP (Zhong et al., 2023) | ICCV'23 | Construct appearance-agnostic motion matrices to decouple motion and appearance. |
| LEO (Wang et al., 2024c) | IJCV'24 | Represent motion as a sequence of flow maps and synthesize videos in the pixel space, and utilize the latent motion diffusion model to model the motion distribution. |
| D-VDM (Shen et al., 2024) | AAAI'24 | Decompose future frames into spatial content and temporal motions. Predict temporal motions based on a 3D-UNet diffusion model. |
| DrNet (Denton et al., 2017) | NeurIPS'17 | Decompose video frames into stationary and time-varying components. |
| DVGPC (Cai et al., 2018) | ECCV'18 | Tackle the severe ill-posedness of human action video prediction with a two-stage framework: generating a human pose sequence from random noise, then creating the human action video. |
| CVP (Ye et al., 2019) | ICCV'19 | Predict the future states of independent entities while reasoning about their interactions, and then synthesize future frames. |
| OCVP-VP (Villar-Corrales et al., 2023) | ICIP'23 | Decouple the processing of temporal dynamics and object interactions. |
| SlotFormer (Wu et al., 2023) | ICLR'23 | Model spatio-temporal relationships by reasoning over object features, and predict accurate future states of objects. |
| OKID (Comas et al., 2023) | L4DC'23 | Decompose a video into moving objects, their attributes and the dynamic modes. |
| MOSO (Sun et al., 2023) | CVPR'23 | Identify motion, scene, and object as the pivotal elements of a video. |

Table 3: Overview of stochastic synthesis methods.

their primary focus on capturing the inherent variability and unpredictability of video sequences. Current research prefers to simulate the complex dynamics of the future using end-to-end generative modeling strategies rather than by introducing stochastic distributions or probabilistic models into deterministic algorithms to obtain diverse outcomes. In the following Section 5, we will discuss generative synthesis approaches, such as diffusion models and auto-regressive models, which are explicitly designed to prioritize the generation of

diverse, high-quality video content. This distinction reflects the evolving focus of video synthesis research, shifting from an emphasis on stochasticity to a broader emphasis on generative capability.

## 4.1 Stochasticity modeling

Uncertain object motion can be modeled either by incorporating stochastic distributions into deterministic frameworks or by directly employing probabilistic models.

**Stochastic distributions.** In the early stages, VPN (Kalchbrenner et al., 2017) employs CNNs to perform multiple predictions in videos based on pixel distributions, while SV2P (Babaeizadeh et al., 2018) enhances an action-conditioned model (Finn et al., 2016) by introducing stochastic distribution estimation. Shifting the focus to a more holistic representation of video elements, the PFP model (Hu et al., 2020) proposes a probabilistic approach for simultaneously synthesizing semantic segmentation, depth maps, and optical flow. Additionally, SRVP (Franceschi et al., 2020) leverages Ordinary Differential Equations (ODEs), while PhyD-Net (Guen & Thome, 2020) employs Partial Differential Equations (PDEs) to model stochastic dynamics. A potential drawback lies in their assumption that physical laws can be linearly disentangled from other factors of variation in the latent space—an assumption that may not hold for all types of videos.

**Probabilistic models.** Building on the pioneering work of Multi-Scale AdvGDL (Mathieu et al., 2016), adversarial training has substantially advanced FFS tasks by improving the prediction of uncertain object motions. Similarly, vRNN (Castrejon et al., 2019) and GHVAE (Wu et al., 2021) enhance VAEs through the incorporation of likelihood networks and hierarchical structures, respectively, thereby contributing a new dimension to the ongoing evolution of stochastic synthesis methods.

To address the challenges of pixel-level synthesis, several studies introduce intermediate representations. S2S (Luc et al., 2017) and Vid2Vid (Wang et al., 2018) incorporate adversarial training into future semantic segmentation frameworks. Additionally, the TPK model (Walker et al., 2017) leverages a VAE to extract human pose information, followed by a GAN to predict future poses and frames. It is worth noting that directly modeling stochastic distributions tends to yield broader predictive coverage but often results in poor visual quality. In contrast, probabilistic models can produce sharper results, but they often face challenges such as mode collapse, training instability, and high computational cost. Bridging these two methodologies, SAVP (Lee et al., 2018) integrates stochastic modeling with adversarial training, achieving a balance between broad predictive diversity and improved visual quality.

Recognizing that object motion is largely deterministic—except in cases of unforeseen events such as collisions, SVG (Denton & Fergus, 2018) models trajectory uncertainty using both fixed and learnable priors, effectively blending deterministic and probabilistic approaches. In a similar vein, but with an emphasis on temporal coherence, Retrospective Cycle GAN (Kwon & Park, 2019) introduces a sequence discriminator to detect fake frames. Building on the paradigm of implicit neural representations (INRs) for video (Sen et al., 2022), this concept of scrutinizing frame authenticity is further extended in DIGAN (Yu et al., 2022), where the focus shifts to a motion discriminator aimed at identifying unnatural motions. StyleGAN-V (Skorokhodov et al., 2022) highlights motion consistency from a different perspective by incorporating continuous motion representations into StyleGAN2 (Karras et al., 2020), enabling consistent generation in high-resolution settings.

**Challenges.** Although stochastic models are capable of capturing a broad spectrum of plausible futures, they often struggle with poor visual quality and increased computational demands. Direct modeling of stochastic distributions often leads to blurred outputs, whereas probabilistic models may encounter issues such as mode collapse and training instability. Striking a balance between diversity, visual fidelity, and computational efficiency remains a significant challenge. Moreover, the assumption that physical laws can be linearly disentangled from other factors of variation may not hold across all types of videos, underscoring the need for more adaptable and generalizable models.

## 4.2 Disentangling components

Stochastic synthesis algorithms primarily focus on modeling the randomness inherent in motion. However, this focus often overlooks the processes of object emergence and disappearance in videos. As a result, many studies isolate motion from other video elements or artificially constrain its evolution, aiming to better understand motion dynamics while reducing the complexity of real-world scenarios.

### 4.2.1 Content and motion

Video synthesis algorithms address the inherent complexity of natural video sequences by emphasizing intricate visual details. To this end, they aim to model appearances through fine-grained local information while simultaneously capturing the dynamic global content of videos. However, in applications such as robotic navigation and autonomous driving, understanding object motion patterns takes precedence over visual fidelity. This shift in priorities has driven the development of algorithms that emphasize object motion prediction and the disentanglement of motion from appearance. A notable early work, CDNA (Finn et al., 2016), sets a precedent by explicitly predicting object motion. It maintains appearance invariance, enabling generalization to unseen objects beyond the training set. MoCoGAN (Tulyakov et al., 2018) learns to disentangle motion from content in an unsupervised manner, and the use of separate encoders for content and motion has since been widely adopted in video prediction models. This concept is further explored in LMC (Lee et al., 2021), where the motion encoder predicts motion based on residual frames, and the content encoder extracts features from the input frame sequence. MMVP (Zhong et al., 2023) takes a different approach by using a single image encoder, followed by a two-stream network that separately handles motion prediction and appearance preservation before decoding. To address the stochastic nature of motion, AMC-GAN (Jang et al., 2018) models multiple plausible outcomes through adversarial training. In contrast, SLAMP (Akan et al., 2021) adopts a non-adversarial approach that focuses on learning stochastic variables for disentangled content and motion representations. Further advancing this line of research, LEO (Wang et al., 2024c) and D-VDM (Shen et al., 2024) leverage diffusion models to achieve more realistic content-motion disentanglement, demonstrating recent progress in this direction.

### 4.2.2 Foreground and background

In future frame prediction, the motion dynamics of foreground objects and background scenes often differ substantially. Foreground objects usually display more dynamic motion, while background scenes tend to remain relatively static. This distinction has motivated research to predict the motion of these components separately, enabling a more nuanced understanding of video dynamics. A notable contribution in this area is DrNet (Denton et al., 2017), which specifically targets scenarios where the background remains largely static across video frames. The model decomposes images into object content and pose, and leverages adversarial training to develop a scene discriminator that determines whether two pose vectors originate from the same video sequence. Similarly, OCVP-VP (Villar-Corrales et al., 2023) employs the slot-wise scene parsing network SAVi (Kipf et al., 2022) to segment scenes hierarchically from the scene level down to individual objects. By focusing on such videos, prediction models can streamline their learning process by eliminating the need to model complex scene dynamics. Both Human-centric tasks, such as predicting human movement and interaction with the environment, and object-centric tasks, such as tracking object motion and positioning, can benefit from this approach.

**Human-centric.** FFS often focuses on foreground motion, especially in scenarios involving complex human movements. A common assumption in these scenarios—reflected across many specialized datasets—is that the background remains relatively static, which is characteristic of datasets focusing on detailed human motion. This has led to a strong research focus on understanding and forecasting human poses to improve foreground motion prediction. A representative example is DVGPC (Cai et al., 2018), which predicts skeleton motion sequences and transforms them into pixel space using a skeleton-to-image Transformer. This method effectively bridges abstract motion representations and the pixel-level demands of video prediction, demonstrating a nuanced understanding of the complexities inherent in human-centric FFS tasks.

**Object-centric.** The concept of object-centric video prediction was first introduced by CVP (Ye et al., 2019), laying the foundation for this specialized subfield of video prediction. SlotFormer (Wu et al., 2023) introduces Transformer-based auto-regressive models to learn object-specific representations from video sequences. This design enables consistent and accurate tracking of individual objects over time. A more recent advancement, OKID (Comas et al., 2023), uniquely decomposes videos into distinct components—specifically, the attributes and trajectory dynamics of moving objects—by employing a Koopman operator. This approach offers a more granular method for analyzing object motion in video sequences, setting it apart from prior methods.

**General.** Methods focused on human poses or objects have shown considerable promise on specific video datasets, but their reliance on predefined structures and limited adaptability to dynamic backgrounds hinder generalization. This limitation is reflected in their performance: while effective under controlled conditions, they often falter when confronted with background variation, revealing insufficient versatility for broader applications. To bridge this gap, MOSO (Sun et al., 2023) proposes a unified framework that identifies motion, scene, and object as the three pivotal elements of a video. It further refines content analysis by distinguishing between scene and object—where the scene denotes the background and the object denotes the foreground—as a finer decomposition of video content. MOSO's core contribution is a two-stage network architecture designed for general-purpose video analysis. In the first stage, the MOSO-VQVAE model encodes video frames into token-level representations, trained via a video reconstruction task to learn informative embeddings. In the second stage, Transformers are employed to handle masked token prediction, enhancing the model's temporal reasoning capabilities. This design enables the model to perform a variety of token-level tasks, including video prediction, interpolation, and unconditional video generation.

**Challenges.** Disentangling content from motion, or foreground from background, in videos is complex due to the intricate interplay between temporal dynamics and spatial context. Models must accurately capture and predict motion patterns, depth, and semantic regions, while maintaining temporal consistency and spatial accuracy. The presence of occlusions, perspective changes, and complex backgrounds further increases the difficulty. The use of high-resolution feature maps and large-scale annotated datasets further exacerbates computational demands. Generalizing to unseen scenes and objects remains a major challenge, requiring robust models capable of adapting to diverse visual appearances and contexts. Applying the concept of separate processing to the generative methods discussed later (in Section 5) also presents a significant challenge.

### 4.3 Motion-controllable synthesis

In the field of FFS, one specialized research direction has emerged that focuses on the explicit control of motion. This approach is distinct in its emphasis on forecasting future object positions based on user-defined instructions, in contrast to the conventional reliance on past motion trends. The central challenge in this domain lies in synthesizing videos that follow these direct instructions while preserving a natural and coherent flow—a task that demands a nuanced understanding of both user intent and motion dynamics within the video context. This challenge underscores the delicate balance between user control and automated imagination, signaling a significant shift in how FFS models are conceptualized and implemented.

**Strokes.** Since there is no historical motion information available for generating videos from one single still image, several methods have emerged that allow for interactive user control. iPOKE (Blattmann et al., 2021) introduces techniques in which local interactive strokes and pokes enable users to deform objects in a still image to generate a sequence of video frames. These strokes represent the user's intended motion for the objects. Building on this innovative direction, the Controllable-Cinemagraphs model (Mahapatra & Kulkarni, 2022) proposes a method for interactively controlling the animation of fluid elements. These advances underscore the growing importance of user-centric approaches in the domain of motion-controllable FFS.

**Instructions.** The integration of instructions across various modalities—including local strokes, sketches, and text—is becoming increasingly common in works aiming to capture user-specified motion trends. Video-

Composer (Wang et al., 2023a) synthesizes videos by combining text descriptions, hand-drawn strokes, and sketches. This approach adheres to textual, spatial, and temporal constraints, leveraging latent video diffusion models and motion vectors to provide explicit dynamic guidance. Essentially, it can generate videos that align with user-defined motion strokes and shape sketches. In a similar vein, DragNUWA (Yin et al., 2023) primarily leverages text for content description and strokes for controlling future motion, enabling the generation of customizable videos. These approaches advance the field of video generation by broadening the spectrum of user input modalities.

**Challenges.** Achieving natural and coherent video synthesis under explicit user control remains a significant challenge. Models must accurately interpret user intent and generate videos that follow specified motion instructions while maintaining both temporal and spatial coherence. Striking a balance between user control and the model's autonomous imagination is essential. Ensuring that the generated videos are both visually compelling and contextually appropriate further increases the complexity, requiring a deep understanding of user intent and video dynamics.

## 5 Generative synthesis

The emergence and disappearance of objects introduce unpredictability, and generative models require a profound understanding of the underlying physical principles that govern the real world. Rather than relying on simplistic linear motion predictions extrapolated from historical frames, they address the challenge using sophisticated and imaginative modeling techniques. As a result, tasks such as transforming one static image into one dynamic video—often referred to as the image animation problem—have emerged as promising candidates for applying generative video prediction techniques. In general, the two mainstream branches are auto-regressive models and diffusion models. Although current auto-regressive models are inferior to diffusion models in terms of image generation quality, their structural consistency with LLMs is highly appealing for building a unified multi-modal system.

### 5.1 Diffusion-based generation

Diffusion models (Ho et al., 2020) have emerged as a dominant approach for image generation. Early attempts at video prediction (Ho et al., 2022; Yang et al., 2023; Harvey et al., 2022; Voleti et al., 2022; Singer et al., 2023), which utilize pixel-space diffusion models by extending the conventional U-Net (Ronneberger et al., 2015) architecture to 3D U-Net structures, are constrained to generating low-resolution and short video clips due to high computational demands. The Latent Diffusion Model (LDM) (Rombach et al., 2022) extends this capability into the latent space of images, significantly enhancing computational efficiency and reducing resource consumption. This advancement has paved the way for applying diffusion models to video generation (Blattmann et al., 2023b).

**Latent diffusion model extensions.** Extensions of LDM have demonstrated strong generative capabilities in video systhesis (Voleti et al., 2022). For instance, Video LDM (Blattmann et al., 2023b) leverages pre-trained image models to generate videos, enabling multi-modal, high-resolution, and long-term video synthesis. Similarly, SEINE (Chen et al., 2024e) introduces a versatile video diffusion model capable of generating transition sequences, thereby extending short clips into longer videos. Burgert et al. (2025) presents a scalable method for finetuning video diffusion models to control motion without modifying model architectures or training pipelines. It introduces a noise warping algorithm that replaces random temporal Gaussian noise with optical flow-guided warped noise, while maintaining spatial Gaussianity. Magic141 (Yi et al., 2025) decomposes the text-to-video generation task into two simpler subtasks for diffusion step distillation: text-to-image generation and image-to-video generation. It shows that image-to-video generation converges more easily than direct text-to-video generation under the same optimization setup, and explores the trade-off between computational cost and video quality.

**Text-guided generation and additional information.** Recent research efforts have focused on harnessing additional modalities alongside RGB images to accomplish the task of text-guided video generation Hong et al. (2022); Singer et al. (2022); Chai et al. (2023). LFDM (Ni et al., 2023) extends latent diffusion models

| Method | Publication | Main Ideas |
|---|---|---|
| MCVD (Voleti et al., 2022) | NeurIPS'22 | Build models from simple non-recurrent 2D convolutional architectures, and train them by randomly masking all past or future frames. |
| Video LDM (Blattmann et al., 2023b) | CVPR'23 | Introduce a temporal dimension into the latent space diffusion model, transforming an image generator into a video generator. |
| SEINE (Chen et al., 2024e) | ICLR'24 | Propose a short-to-long video diffusion model, aimed at generating coherent long videos at the "story-level". |
| Go-with-the-Flow (Burgert et al., 2025) | CVPR'25 | Achieve user-friendly motion control through structured latent noise sampling derived from optical flow fields. |
| Magic141 (Yi et al., 2025) | CVPR'25 | Explore rapid video generation by decomposing the text-to-video generation task into two subtasks for diffusion step distillation. |
| LFDM (Ni et al., 2023) | CVPR'23 | Synthesize an optical flow sequence in the latent space based on given conditions to warp the input image. |
| Seer (Gu et al., 2024) | ICLR'24 | Inflate a pre-trained T2I model along the temporal axis, and integrate global sentence-level instructions into each generated frame. |
| Emu Video (Girdhar et al., 2024) | ECCV'24 | Investigate key design aspects of text-to-image-to-video generative models, including noise scheduling for diffusion and multi-stage training. |
| DynamiCrafter (Xing et al., 2024) | ECCV'24 | Incorporate image guidance into the text-to-video diffusion model, by projecting images into a text-aligned context space. |
| SparseCtrl (Guo et al., 2024) | ECCV'24 | Keep the pre-trained T2V model unchanged and introduce an additional condition encoder to process input like sketches, depth maps, and RGB keyframes. |
| PEEKABOO (Jain et al., 2024) | CVPR'24 | Integrate user-interactive control into the T2V model via a masked attention module, without requiring extra training or inference overhead. |
| VACE (Jiang et al., 2025) | preprint | Organizing video task inputs into a unified interface to achieve a unified approach, involves injecting different task concepts into the video synthesis model. |
| CineMaster (Wang et al., 2025a) | SIGGRAPH'25 | A framework for 3D-aware and controllable text-to-video generation, empowering users with precise control over the scene. |
| SkyReels-A2 (Fei et al., 2025) | preprint | Introduce the novel elements-to-video task, generating diverse and high-quality videos with precise element control. |
| MicroCinema (Wang et al., 2024b) | CVPR'24 | Use the Appearance Injection Network to enhance appearance preservation. Employ an Appearance Noise Prior to retain the capabilities of pre-trained models. |
| LivePhoto (Chen et al., 2024d) | ECCV'24 | Enable users to control the temporal motion of images with text, and decode motion-related instructions into videos using a generator equipped with a motion module. |
| I2VGen-XL (Zhang et al., 2023) | *preprint* | Propose a cascaded approach to decouple the two factors of semantics and quality, enhancing details through optimization with high-quality data. |
| SkyReels-V2 (Chen et al., 2025) | preprint | Integrating multi-modal LLM, conducting progressive-resolution pretraining, achieving realistic long-form video synthesis and professional film-style generation. |
| Art-v (Weng et al., 2024) | CVPRW'24 | Generate frames conditioned on the previous one auto-regressively. Maintain the capabilities of the pre-trained model without modeling long-range motions. |
| GAIA-1 (Hu et al., 2023a) | *preprint* | Cast world modeling as an unsupervised sequence modeling problem by transforming text, video, and ego-vehicle actions into discrete tokens. |
| Video Transformer (Weissenborn et al., 2020) | ICLR'20 | Study how pure Transformer-based video classification models extract and encode long spatio-temporal token sequences, outperforming previous 3D CNNs. |
| LVT (Rakhimov et al., 2021) | VISIGRAPP'21 | Reduce the computational requirements of video generative models by modeling dynamics in the latent space. |
| Nuwa (Wu et al., 2022a) | ECCV'22 | A unified multimodal model that handles language, images, and videos, employing a 3D Nearby Attention mechanism to reduce computational complexity. |
| Nuwa-Infinity (Liang et al., 2022) | NeurIPS'22 | For variable-length generation tasks targeting images of arbitrary sizes or long-duration videos, introduce a global patch-based and a local visual token-based auto-regressive model. |
| MMVG (Fu et al., 2023) | CVPR'23 | Discretize video frames into visual tokens and propose a multimodal masked video generation approach to tackle the text-guided video completion task. |
| MAGVIT (Yu et al., 2023b) | CVPR'23 | Introduce a 3D tokenizer to quantize videos into spatio-temporal visual tokens, and propose an embedding strategy for masked video token modeling. |
| LVM (Bai et al., 2024) | CVPR'24 | Represent diverse visual data as sequences of discrete tokens, and demonstrate that visual auto-regressive models can scale effectively. |
| Painter (Wang et al., 2023b) | CVPR'23 | Specifying both the input prompts and outputs of visual tasks as images, and performing standard masked image modeling on the stitch of input and output image pairs. |
| MAGVIT-v2 (Yu et al., 2024) | ICLR'24 | Explore the design of discrete token tokenizers to boost auto-regressive models, and introduce a lookup-free quantizer to enhance video tokenizers. |
| VideoPoet (Kondratyuk et al., 2024) | ICML'24 | Incorporate multimodal generative objectives—including images, videos, text, and audio—within an auto-regressive Transformer framework. |

Table 4: Overview of generative synthesis methods.

to synthesize optical flow sequences in the latent space based on textual guidance. Seer (Gu et al., 2024) inflates Stable Diffusion (Rombach et al., 2022) along the temporal axis, enabling the model to utilize natural language instructions and reference frames to envision multiple variations of future outcomes. Emu Video (Girdhar et al., 2024) generates an image conditioned on textual guidance and extrapolates it into a

video, making it adaptable to diverse textual inputs. DynamiCrafter (Xing et al., 2024) extends text-guided image animation to open-domain image scenarios. SparseCtrl (Guo et al., 2024) supports sketch-to-video generation, depth-to-video generation, and video prediction with an expanded range of input modalities. Other methods, such as PEEKABOO (Jain et al., 2024), explore interactive synthesis, aiming to unlock unprecedented applications and creative potential. VACE (Jiang et al., 2025) proposes a unified conditioning format called the video condition unit, which encodes text instructions, reference images, and masks. It incorporates a context adapter module that injects task-specific context into the model using structured temporal and spatial representations. This design ensures coherence across frames and spatial regions during generation and supports diverse applications through flexible task composition. SkyReels-A2 (Fei et al., 2025) defines the elements-to-video task, which involves generating videos by integrating arbitrary visual elements such as characters, objects, and backgrounds through reference images under the guidance of text prompts. The model demonstrates a strong ability to preserve the distinct visual details of each individual element throughout the generation process while simultaneously producing results that are coherently blended. This capability represents a significant step forward in the field by shifting the focus of video synthesis from single-condition generation to multi-condition combinatorial control. CineMaster (Wang et al., 2025a) proposes a 3D-aware, controllable text-to-video framework where users interactively define 3D control signals such as object bounding boxes and camera trajectories to guide a text-to-video diffusion model. To address the scarcity of annotated data, it introduces an automated pipeline that extracts 3D control signals from large-scale video datasets.

**Preserving text guidance.** Some works aim to achieve a more precise interpretation of textual guidance and preserve this information across the temporal dimension. MicroCinema (Wang et al., 2024b) adopts a divide-and-conquer strategy to address challenges related to appearance and temporal coherence. It employs a two-stage generation pipeline, first creating an initial image using an existing text-to-image generator, and then introducing a dedicated text-guided video generation framework for motion modeling. LivePhoto (Chen et al., 2024d) proposes a framework that incorporates motion intensity as an auxiliary factor to enhance control over motion dynamics. It further introduces a text re-weighting mechanism to emphasize motion-related descriptions, demonstrating strong performance in text-guided video synthesis tasks. I2VGen-XL (Zhang et al., 2023) utilizes static images to provide semantic and quality-related guidance, highlighting the diversity of approaches in text-guided video synthesis. SkyReels-V2 (Chen et al., 2025) introduces a multi-modal LLM to interpret narrative semantics by decomposing script-like prompts into shot-level descriptions and utilizes a specialized video captioning module to automatically annotate training data with rich cinematographic language. This methodology enables its 14B parameter model to generate videos exceeding 30 seconds in duration while exhibiting a professional cinematic style, representing the first open-source model to achieve long-form video synthesis that closely approximates the quality and coherence of film production.

**Integrating auto-regressive models.** While the dominance of diffusion models in generative tasks is rising, some researchers still attempt to preserve the architecture of auto-regressive models (Weng et al., 2024), or develop new diffusion architectures to integrate the strengths of both paradigms (Chen et al., 2024a). Early generative video synthesis algorithms are constrained by limited data availability and model scalability, yet they lay the groundwork for integrating LLMs and diffusion models in video generation. VDT (Lu et al., 2024) represents one of the earliest efforts to incorporate a Transformer-based backbone into diffusion models for video generation. Inspired by the success of DiT (Peebles & Xie, 2023) in image synthesis, VDT replaces the conventional U-Net with a spatio-temporal Transformer that explicitly models both spatial and temporal dependencies. W.A.L.T. (Gupta et al., 2025) provides the first successful empirical demonstration of a Transformer-based backbone for jointly training image and video latent diffusion models. Recently, GAIA-1 (Hu et al., 2023a) and Sora (Brooks et al., 2024) have also leveraged the strengths of DiT to enable more creative, generalizable, and scalable video synthesis. RIVER (Davtyan et al., 2023) leverages flow matching (Lipman et al., 2023) for efficient video prediction by conditioning on a small set of past frames in the latent space of a pre-trained VQGAN (Esser et al., 2021). Collectively, these advances underscore the potential of diffusion models to produce high-quality, controllable, and diverse video content, thereby pushing the boundaries of what is achievable in computer vision and artificial intelligence.

| Method | Model Size |
|---|---|
| HunyuanVideo (Kong et al., 2024) | 13B |
| Step-Video-T2V (Ma et al., 2025) | 30B |
| Step-Video-TI2V (Huang et al., 2025) | 30B |
| Seaweed-7B (Seawead et al., 2025) | 7B |
| MAGI-1 (Sand-AI, 2025) | 24B |

Table 5: Recently open-sourced large models for video generation.

**Recent open-sourced solution.** HunyuanVideo (Kong et al., 2024) is a very large open-source video generative model (13B parameters). Relying on key strategies such as data curation and image-video co-training, the model reaches a level of generation quality comparable to or even better than that of the leading closed-source models, which effectively narrows the performance gap between open-source and closed-source solutions. Ma et al. (2025) release an even larger video generative model, Step-Video-T2V with 30B parameters, and a new benchmark for video generation, Step-Video-T2V-Eval. The model incorporates a deep compression video Variational Autoencoder for compact and efficient video representation, and leverages a flow matching approach to train a DiT (Peebles & Xie, 2022) architecture for denoising input noises into latent frames. To enhance the visual quality and mitigate artifacts, a video-based direct preference optimization method is applied. For the task of text-driven image-to-video generation, Huang et al. (2025) release Step-Video-TI2V, a model of equal scale with 30B parameters, along with its corresponding benchmark, Step-Video-TI2V-Eval. It is noteworthy that improving video generation performance does not necessarily require increasing the number of model parameters. Seaweed-7B (Seawead et al., 2025), with only 7B parameters, demonstrates generation quality comparable to that of models with tens of billions of parameters. This provides important insights into how the performance of medium-sized DiT models can be effectively enhanced without relying solely on scaling up model size. Sand-AI (2025) has open-sourced a 24B parameter video generative model, MAGI-1, that divides video frame sequences into fixed-length video chunks and adopts a chunk-wise auto-regressive diffusion framework, facilitating text-driven image-to-video generation through chunk-wise prompting. This design illustrates a promising direction in which high-fidelity video synthesis and fine-grained instruction control can be unified.

**Challenges.** While diffusion models have made significant strides in video generation, several critical challenges remain. Ensuring temporal coherence and consistency across frames is essential for achieving realism, yet remains a major challenge (Chen et al., 2024f; Xu et al., 2024). The computational efficiency and scalability of diffusion models are hindered by their resource-intensive nature, which limits their broader adoption (Peebles & Xie, 2023). Issues of controllability and interpretability persist, as textual guidance does not always align with visual outcomes, and model behaviors often remain opaque. Data availability and diversity are critical for training robust models, but acquiring comprehensive and diverse datasets remains a major bottleneck.

## 5.2 Token-based generation

Diffusion-based methods have garnered significant attention in the realm of image and video generation. However, these models typically have smaller parameter scales compared to contemporary large-scale language models. Recent research has increasingly focused on exploring how LLMs can be employed for such tasks, leveraging their optimization techniques and accumulated insights to investigate the applicability of scaling laws in the visual domain.

**Key components.** Implementing token-based FFS requires two key components: a high-quality visual tokenizer and an efficiently scalable LLM framework. Innovations such as VQ-VAE (Van Den Oord et al., 2017) and VQGAN (Esser et al., 2021) integrate auto-regressive models with adversarial training strategies to tackle image quantization and tokenization. An effective visual tokenizer should minimize the number of tokens per image or video clip segment while preserving near-lossless visual fidelity. However, the substantial

token requirements for lossless reconstruction—particularly for high-resolution images—present challenges in processing long video sequences during training, thereby limiting video generation capabilities.

**Latent space modeling.** Even before the advent of LLMs (Brown et al., 2020; Achiam et al., 2023), Transformers have already made a significant impact on time-series modeling. Video Transformer (Weissenborn et al., 2020) pioneer the use of Transformer architectures in video synthesis by introducing an auto-regressive modeling approach. Despite its success, it inherits common limitations of Transformer-based models, including high training resource demands and slow inference speed. The Latent Video Transformer (LVT) (Rakhimov et al., 2021) introduces a novel latent-space approach that auto-regressively models temporal dynamics and predicts future features, significantly reducing computational overhead. Other works, such as VideoGen (Zhang et al., 2020c) and Video VQ-VAE (Walker et al., 2021), also leverage the VQ-VAE framework (Van Den Oord et al., 2017) to extract discrete tokens for video prediction. In contrast, Phenaki (Villegas et al., 2022) improves upon the ViViT (Arnab et al., 2021) architecture to extract continuous tokens. The NUWA framework (Wu et al., 2022a) proposes a versatile 3D Transformer-based encoder-decoder architecture that is adaptable to diverse data modalities and tasks, further demonstrating the potential of Transformers in video synthesis. NUWA-Infinity (Liang et al., 2022) builds upon this with an innovative generation mechanism designed to enable infinite high-resolution video synthesis, reflecting ongoing efforts to unify generative tasks across modalities.

**In-context learning in visual domain.** LVM (Bai et al., 2024) introduces sequential modeling to enhance the learning capacity of large-scale vision models, demonstrating the scalability and flexibility of sequence-based models in in-context learning. The concept of "visual sentence" is proposed, in which a sequence of intrinsically related images is organized analogously to a linguistic sentence. This allows the model to leverage sequential information for sentence continuation and other visual tasks without relying on non-pixel-level knowledge. Painter (Wang et al., 2023b) proposes a general framework for visual learning that enables images to "speak" through in-context visual understanding, thereby enhancing both image generation and interpretation. SegGPT (Wang et al., 2023c) explores the use of a GPT-based architecture for image segmentation, introducing the concept of "segmenting everything" and demonstrating the potential for unified segmentation under unsupervised learning settings, thereby advancing generalization in visual segmentation tasks.

**Advances in video generation.** In the domain of video generation, MAGVIT (Yu et al., 2023b) presents a masked generative video Transformer that efficiently processes videos by masking certain regions and predicting the missing segments. MAGVIT-v2 (Yu et al., 2024) suggests that Transformer-based models may surpass diffusion models in visual generation tasks, highlighting the pivotal role of visual tokenizers. VideoPoet (Kondratyuk et al., 2024) introduces a large language model for zero-shot video generation, pushing the boundaries of unsupervised video synthesis. It enables users to generate or edit videos based on high-level textual prompts and excels at capturing temporal and contextual relationships within video data.

Text-guided generative video synthesis algorithms generate sequences of frames by integrating contextual visual information with textual guidance. The Text-guided Video Completion (TVC) task entails completing videos under various conditions, including the first frame (video prediction), the last frame (video rewind), or both (video transition), guided by textual instructions. MMVG (Fu et al., 2023) addresses the TVC task using an auto-regressive encoder-decoder architecture that integrates textual and visual features, forming a unified framework capable of handling diverse video synthesis tasks.

Collectively, these studies integrate visual tokenizers and LLMs to establish unified and scalable frameworks for visual learning, thereby driving significant progress in FFS tasks. The evolving landscape of FFS research continues to highlight the potential of Transformers and LLMs in unifying generative tasks across modalities.

**Challenges.** Token-based generation for FFS faces significant challenges, particularly in designing efficient visual tokenizers that balance a minimal number of tokens with near-lossless reconstruction, especially for high-resolution content. The high computational demands of Transformer-based models also present barriers to adoption, particularly in resource-constrained environments. When computational resources are limited, the emergence phenomena reported in previous studies under this paradigm may fail to manifest (Bai et al.,

2024). Recent work (Sun et al., 2024a) suggests that one reason token-based models struggle to match the visual quality of diffusion models is their limited integration with high-quality community assets, such as robust training infrastructure and curated datasets.

## 6 Application Realms

The applications of FFS are extensive and diverse, permeating numerous domains and highlighting its increasing importance as a versatile tool. Across these domains, FFS serves as a fundamental enabler for predictive modeling and decision-making, allowing systems to anticipate future states based on past observations. This predictive capability is crucial for enhancing safety, efficiency, and adaptability in dynamic environments.

**World model.** World models (Ha & Schmidhuber, 2018b;a; Zhu et al., 2024) provide a general-purpose framework for simulating and predicting the dynamics of complex systems. These models are widely employed in reinforcement learning and robotics, enabling agents to make informed decisions and perform actions that lead to desired outcomes. FFS serves as a key learning objective in the development of world models (Hafner et al., 2020; 2021; 2023; Wang et al., 2024a; Ge et al., 2024; Agarwal et al., 2025). Recent work (Escontrela et al., 2024) demonstrates that video prediction can also be incorporated into reward modeling to further support reinforcement learning. GameNGen (Valevski et al., 2025) has demonstrated exceptional world modeling performance and strong instruction-following ability in controllable FFS.

**Autonomous driving.** FFS is indispensable for autonomous vehicles, including self-driving cars and drones, as it enables them to anticipate the movement of objects, pedestrians, and other vehicles. This predictive capability is critical for ensuring safe and efficient navigation. For instance, GAIA-1 (Hu et al., 2023a) employs a unified world model that integrates multimodal LLMs and diffusion processes to predict control signals and future frames, thereby enhancing decision-making capabilities in autonomous systems. In most existing driver-assistance systems, visual input is first transformed into structured representations—such as objects, lane markings, and traffic lights—followed by downstream predictions based on these modalities. Effectively leveraging raw visual information for trajectory prediction in real-world scenarios remains a significant challenge (Nayakanti et al., 2023; Varadarajan et al., 2022). Most existing trajectory prediction approaches rely solely on historical detection records of surrounding vehicles and pedestrians. Recent studies (Gu et al., 2023) have demonstrated that fully exploiting semantic information from visual inputs can significantly improve behavior prediction in dynamic scenes.

**Robotics.** In the field of robotics, FFS is employed to guide robotic agents through dynamic environments. It enables them to effectively plan paths, manipulate objects, and avoid obstacles, as demonstrated by (Finn & Levine, 2017). By predicting future states, robotic systems can make proactive decisions, thereby enhancing adaptability and operational efficiency in complex environments. The GR-1 and GR-2 approaches (Wu et al., 2024a; Cheang et al., 2024) demonstrate that visual robotic manipulation can benefit significantly from large-scale video generative pre-training. After being pre-trained on large-scale video datasets, GR-1/2 can be seamlessly fine-tuned on robot-specific data, exhibiting strong generalization to unseen scenes and objects.

**Film production.** FFS has found valuable applications in the film industry, particularly in special effects, animation, and pre-visualization. It assists filmmakers in generating realistic scenes and enhancing the overall cinematic experience. For example, Mahapatra & Kulkarni (2022) utilize FFS to generate visually compelling sequences that enhance narrative coherence and support artistic expression in filmmaking.

**Meteorological.** FFS plays a vital role in weather forecasting by assisting meteorologists in simulating and predicting atmospheric dynamics. By accurately forecasting future spatio-temporal patterns, FFS enhances the precision of weather prediction models, as demonstrated by Shi et al. (2017). This capability is essential for both operational weather forecasting and disaster preparedness.

**Anomaly detection.** Liu et al. (2018) propose a video anomaly detection approach based on future frame prediction, under the assumption that normal events are predictable whereas abnormal events deviate from expected patterns. The method introduces a motion constraint alongside appearance constraints to ensure that the predicted future frames align with the ground truth in both spatial and temporal dimensions.

Overall, these diverse applications underscore the significance and broad potential of FFS as a powerful tool for understanding and interacting with the dynamic world. Its capacity to predict future states from past observations positions it as a valuable asset across a wide spectrum of domains, including artificial intelligence, robotics, entertainment, and beyond.

**Challenges.** When applying FFS methods to real-world scenarios, several potential challenges may arise. For instance, domain-specific applications may require solutions for few-shot learning (Gui et al., 2018) or test-time adaptation (Choi et al., 2021). Another challenge lies in the effective integration of vectorized data with image-based inputs. Moreover, interpretability remains a concern in real-world applications, as end-to-end FFS methods often lack transparency and are difficult to interpret.

# 7 Related Work

In the related work section, we delineate our survey's unique focus on FFS by comparing it with existing literature on video prediction and video diffusion models, highlighting both similarities and key distinctions.

**Previous surveys on video prediction.** The field of video prediction, along with related areas such as action recognition and spatio-temporal predictive learning, has witnessed significant progress in recent years, largely driven by deep learning techniques. Several comprehensive surveys have provided overviews of state-of-the-art methods, benchmark datasets, and evaluation protocols in this domain. Zhou et al. (2020) review next-frame prediction models developed prior to 2020, categorizing them into sequence-to-one and sequence-to-sequence architectures. The survey compares these approaches by analyzing their architectural designs and loss functions, and provides quantitative performance comparisons based on standard datasets and evaluation metrics. Oprea et al. (2020) present a comprehensive review of deep learning methods for video prediction, outlining fundamental concepts and analyzing existing models based on a proposed taxonomy. The survey also includes experimental results to enable quantitative assessment of the state-of-the-art. Rasouli (2020) provide an overview of vision-based prediction algorithms with a focus on deep learning approaches. They categorize prediction tasks into video prediction, action prediction, trajectory prediction, body motion prediction, and other related applications, and discuss common network architectures, training strategies, data modalities, evaluation metrics, and benchmark datasets. While these surveys focus on early technical developments in video prediction, our survey specifically emphasizes the synthesis aspect of FFS and its evolution towards generative methodologies, highlighting the rising significance of generative models.

Kong & Fu (2022) survey state-of-the-art techniques in action recognition and prediction, covering existing models, representative algorithms, technical challenges, action datasets, evaluation protocols, and future research directions. Tan et al. (2023) introduce OpenSTL, a unified benchmark for spatio-temporal predictive learning, which categorizes methods into recurrent-based and recurrent-free models. The paper provides standardized evaluations on multiple datasets and offers an in-depth analysis of how model architecture and dataset characteristics influence performance. For our survey, these reviews are excellent supplements regarding technical details such as architectures and benchmarks.

**Surveys on video diffusion models.** More recently, the emergence of video diffusion models has led to several specialized surveys. Xing et al. (2023) present a comprehensive review of video diffusion models in the era of AI-generated content, categorizing existing works into video generation, video editing, and other video understanding tasks. The survey provides an in-depth analysis of the literature in these areas and discusses current challenges and future research trends. Li et al. (2024) survey recent advances in long video generation, summarizing existing methods into two key paradigms: divide-and-conquer and temporal auto-regressive modeling. They also provide a comprehensive overview and categorization of datasets and evaluation metrics, and discuss emerging challenges and future directions in this rapidly evolving field. Sun et al. (2024b) review Sora, OpenAI's text-to-video model, categorizing the related literature into three themes—evolutionary

generators, excellence in pursuit, and realistic panoramas—while also discussing datasets, evaluation metrics, existing challenges, and future directions. Complementarily, Liu et al. (2024b) provide a comprehensive analysis of Sora's underlying technologies, system design, current limitations, and its potential role in the broader landscape of large vision models. Melnik et al. (2024) offer an in-depth exploration of the critical components of video diffusion models, focusing on their applications, architectural design, and temporal dynamics modeling. These surveys offer highly specific analyses of recent models. In contrast, our survey provides a broader review of the FFS methodology, positioning models like Sora as examples within the evolving landscape of the generative approaches to FFS.

**Our survey focus.** Our survey provides a comprehensive review of both historical and recent works in FFS, with a particular focus on the transition from deterministic to generative synthesis methodologies. The survey highlights key advances and methodological shifts, emphasizing the growing role of generative models in producing realistic and diverse future frame predictions.

# 8 Conclusion

In this survey, we have systematically charted the evolution of Future Frame Synthesis (FFS), tracing its journey from deterministic prediction to the era of large-scale generative models. Our taxonomy, organized by the degree of algorithmic stochasticity, not only categorizes the field but also illuminates the fundamental methodological shifts that have defined its progress. Looking forward, we argue that the trajectory of FFS is not monolithic but is bifurcating into two distinct, yet potentially synergistic, research frontiers, each with its own set of grand challenges and open questions.

**The First Frontier**: Pragmatic Real-Time Synthesis. This direction prioritizes efficiency, precision, and applicability for high-definition, low-latency tasks. Rather than pursuing general world knowledge, its goal is to perfect low-level objectives like video compression, frame interpolation, and short-term forecasting. Key open questions in this domain include:

Hybrid Model Architectures: Can we design lightweight, deterministic modules—for instance, advanced optical flow or motion field estimators—that can be efficiently "plugged into" large generative models? Such hybrids could leverage the generative prior for context while using the deterministic module to enforce short-term, pixel-perfect motion accuracy at a low computational cost.

Knowledge Distillation for Motion: How can the rich, implicit motion understanding learned by massive models like SVD (Blattmann et al., 2023a) or Sora (Brooks et al., 2024) be effectively distilled into compact, recurrent-free networks? Developing novel distillation techniques tailored for temporal dynamics, rather than just static image features, is critical for real-world deployment on edge devices.

**The Second Frontier**: Generative World Simulation. This path is more ambitious, aiming to develop models with a fundamental understanding of the physical world, capable of generating diverse, coherent, and long-duration videos by leveraging vast computational and data resources. This research directly contributes to the long-term vision of creating general-purpose world models. The most pressing challenges here are not just about scaling, but about fundamental model capabilities:

Bridging the Paradigm Gap: How do we unify the strengths of disparate synthesis paradigms? Current methods excel at either precise motion warping (motion-based synthesis) or creative content generation (diffusion models), but rarely both. A pivotal breakthrough would be a unified framework that generates novel objects and events while adhering strictly to the predicted trajectories of existing elements.

Solving the Tokenizer Bottleneck: Is the persistent quality gap between token-based generative models and their diffusion-based counterparts a fundamental limitation of processing visual data as discrete sequences, or is it an engineering problem of tokenizer design and training infrastructure? Exploring hybrid auto-regressive/diffusion backbones or developing novel, lookup-free quantization schemes could unlock the scalability of LLM-like architectures for high-fidelity video generation.

From Semantic to Physical Control: Current controllability is largely limited to semantic instructions like text or sparse inputs like strokes. The next leap is to enable fine-grained, quantitative control over the

underlying physics of a scene. Can we instruct a model to generate a video where "the ball has a mass of 5kg and a friction coefficient of 0.3," and see a physically plausible result? This requires moving beyond pattern recognition to building genuine, albeit simplified, simulation engines.

Progress in this second frontier is fundamentally gated by the evolution of our evaluation metrics. Metrics must evolve to reward not just perceptual quality, but also long-term temporal coherence, causal reasoning, and adherence to physical principles. The ultimate objective is to transform FFS from a task of visual pattern continuation into one of dynamic world simulation. As we have detailed, achieving this vision requires a concerted effort across model architecture, data curation, and evaluation philosophy. We anticipate that the insights and categorization provided in this survey will serve as a valuable roadmap for researchers navigating this exciting and rapidly changing domain.

**Broader impact.** The performance improvements of future prediction models that do not incorporate generative content do not bring significant negative social impacts. With the rise of AI-generation methods, the way we create and consume video content is rapidly evolving. While AI-generated technologies hold the potential to save substantial time for content marketers and video creators, they also raise numerous ethical concerns. On the positive side, AI video generation can save both time and money for content creators, expand their imagination, and democratize video creation, making it easier for small businesses and individuals to compete with companies that have large marketing budgets. However, it's inevitable that distinguishing between real and AI-generated content will become increasingly challenging. Without proper safeguards, there is a risk of misinformation, propaganda, and the manipulation of public opinion. Generative videos might misrepresent people and events, reflect biases and discrimination from the training set, pose legal and copyright issues, and potentially lead to job losses for human video editors and animators. Apart from imposing constraints on content generation service providers, such as preventing inappropriate generation and adding watermarks, the exploration of AI-Generated Media Detection methods is also very useful. To mitigate risks, content generation service providers can prevent inappropriate generation and add watermarks. Exploring methodologies for detecting AI-generated media (Zou et al., 2025; Chen et al., 2024b) holds significant value as well.

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
