# OpenReview forum: "A Survey on Future Frame Synthesis: Bridging Deterministic and Generative Approaches"
_TMLR — Accepted by TMLR_

### Review · Reviewer_H6Vr · 2025-04-28

**Summary Of Contributions:**

This paper presents a survey on the Future Frame Synthesis (FFS) problem, which focuses on generating future frame sequences conditioned on existing content. It reviews relevant datasets and methods—both deterministic and generative—to provide a comprehensive overview of the field.

**Audience:**

Yes

**Claims And Evidence:**

Yes

**Requested Changes:**

1. It would be helpful to add the recent survey paper on video diffusion models [1] to provide a more comprehensive overview.

2. I suggest including more recent works on video generation evaluation [2–3] in Section 2.3.1. Although they are not specifically designed for FFS, comparisons and discussions with them could enrich the survey and offer valuable perspectives.

3. Recent works have attempted to establish connections between neural video synthesis and 3D representation learning (such as 3DGS) [4–5]. It may be interesting to discuss whether these developments can offer any insights for the FFS task.

4. It would strengthen the paper to include a dedicated section discussing future trends and highlighting some of the most promising directions based on the current progress.

5. For consistency, please clarify sub-subsection titles by removing the periods at the end (e.g., changing "3.1.1 Recurrent networks." to "3.1.1 Recurrent networks") to maintain a uniform style throughout the paper.

6. Consider adding more figures to illustrate the problem formulation, representative datasets, and task pipelines, which could help improve the paper’s readability. There is not any figure in the paper yet.

7. While involving stochasticity in modeling approaches is an important shift, it does not necessarily imply a method is truly "generative," where the goal is typically to maximize the likelihood of the observed data samples. It might be helpful to clarify this distinction more explicitly in the paper.


Ref: \
[1] Melnik A, Ljubljanac M, Lu C, Yan Q, Ren W, Ritter H. Video diffusion models: A survey. arXiv preprint arXiv:2405.03150. 2024 May 6. \
[2] Huang Z, He Y, Yu J, Zhang F, Si C, Jiang Y, Zhang Y, Wu T, Jin Q, Chanpaisit N, Wang Y. Vbench: Comprehensive benchmark suite for video generative models. InProceedings of the IEEE/CVF Conference on Computer Vision and Pattern Recognition 2024 (pp. 21807-21818). \
[3] Liu J, Qu Y, Yan Q, Zeng X, Wang L, Liao R. Fr\'echet Video Motion Distance: A Metric for Evaluating Motion Consistency in Videos. arXiv preprint arXiv:2407.16124. 2024 Jul 23. \
[4] Lei J, Weng Y, Harley A, Guibas L, Daniilidis K. Mosca: Dynamic gaussian fusion from casual videos via 4d motion scaffolds. arXiv preprint arXiv:2405.17421. 2024 May 27. \
[5] Chen Z, Wang Y, Wang F, Wang Z, Liu H. V3d: Video diffusion models are effective 3d generators. arXiv preprint arXiv:2403.06738. 2024 Mar 11. \

**Strengths And Weaknesses:**

Strengths:
* The technical language and descriptions are generally clear and easy to follow.

* The discussion on the methodological shift from deterministic to generative approaches is particularly interesting and provides valuable insights.

Weaknesses: please see the comments below.

---

> ### Author Response · Authors · 2025-05-22
> **Response to Reviewer H6Vr**
>
> Dear Reviewer H6Vr,
>
> We sincerely appreciate your precious time and constructive comments. Your suggestions regarding literature integration and terminology clarification significantly strengthened our survey. The changes can be seen in the newest version of the paper, where they are highlighted in blue. In the following, we would like to answer your concerns separately.
>
> **Weakness 1**: It would be helpful to add the recent survey paper on video diffusion models [1] to provide a more comprehensive overview.
>
> **Response**: Thank you for the suggestion. We have included the survey [1] in `Section 7`. We briefly summarize its contributions, including the design of video diffusion architectures and temporal dynamics modelling.
>
> **Weakness 2**: I suggest including more recent works on video generation evaluation [2–3] in Section 2.3.1. Although they are not specifically designed for FFS, comparisons and discussions with them could enrich the survey and offer valuable perspectives.
>
> **Response**: We have incorporated both VBench [2] and Fréchet Video Motion Distance [3] in `Section 2.3.1`. We discuss how these evaluation metrics enhance existing benchmarks by targeting motion consistency and generation diversity.
>
> **Weakness 3**: Recent works have attempted to establish connections between neural video synthesis and 3D representation learning (such as 3DGS) [4–5]. It may be interesting to discuss whether these developments can offer any insights for the FFS task.
>
> **Response**: We have added a new paragraph in `Section 3.1.3`, discussing how concepts from 3D representation learning can contribute to FFS [4-5]. We highlight their potential to improve motion representation and camera-consistent generation, which may help address key limitations in current methods.
>
> **Weakness 4**: It would strengthen the paper to include a dedicated section discussing future trends and highlighting some of the most promising directions based on the current progress.
>
> **Response**: We have expanded `Section 2.3` with development trends, which comprehensively analyzes several future directions and open challenges in the FFS field. For example, we highlight the growing shift from low-level pixel-based metrics (e.g., PSNR, SSIM) to holistic evaluation paradigms such as VBench [2] and FVMD [3], which better reflect perceptual quality, temporal consistency, and real-world reasoning. We also discuss the challenges in long-term video synthesis, emphasizing the need for models that can maintain object consistency over extended time horizons and leverage the priors of large-scale diffusion models.
>
> **Weakness 5**: For consistency, please clarify sub-subsection titles by removing the periods at the end (e.g., changing "3.1.1 Recurrent networks." to "3.1.1 Recurrent networks") to maintain a uniform style throughout the paper.
>
> **Response**: We have unified all titles across the paper by removing the redundant periods. Thank you for catching this inconsistency.
>
> **Weakness 6**: Consider adding more figures to illustrate the problem formulation, representative datasets, and task pipelines, which could help improve the paper’s readability. There is not any figure in the paper yet.
>
> **Response**: We have added a figure showcasing diverse datasets to highlight the recent trend toward higher video complexity and resolution.
>
> **Weakness 7**: While involving stochasticity in modeling approaches is an important shift, it does not necessarily imply a method is truly "generative," where the goal is typically to maximize the likelihood of the observed data samples. It might be helpful to clarify this distinction more explicitly in the paper.
>
> **Response**: Thank you for highlighting this ambiguity. At the end of `Section 2.2`, we clarify that although stochastic models inject randomness (e.g., via latent sampling), they may not explicitly maximize data likelihood or learn the underlying data distribution. We contrast this with generative models, which are trained for full sample synthesis, and we explain how both types fit into our categorization scheme.

---

> > ### Comment · Reviewer_H6Vr · 2025-05-25
> >
> > Thanks for your detailed response. I don't have any more concerns and recommend accepting this paper.

---

### Review · Reviewer_HFqA · 2025-05-05

**Summary Of Contributions:**

The paper is a survey that covers future frame synthesis (FFS). Several surveys exist in the space of future frame synthesis, however, this survey focuses on framing algorithms and approaches based on how well they cover stochasticity. After defining the FFS problem and using a broad definition that incorporates conditioning on human guidance and non-human created auxiliary data. The survey breaks the problem of future frame synthesis by deterministic, stochastic, and generative approaches. Each of the three approaches is then further broken down by more refined methods types, and the challenges of each are discussed. Beyond the summary of the field provided, the contribution is provided in the conclusion where the recommendation to split FFS research into two subfields is provided. One to focus on high-definition video applications, and the second developing algorithms that learn to appropriately model physics and the world.

**Audience:**

Yes

**Broader Impact Concerns:**

I have no concerns about broader impact.

**Claims And Evidence:**

Yes

**Requested Changes:**

None

**Strengths And Weaknesses:**

Strengths
- The paper is well written and easy to read.
- The break down of the algorithms by stochasticity is intuitive and easy to follow.
- The challenges sections associated with each method discussed highlights and makes clear where the field is.

Weaknesses
- I am not sure how novel the stochasticity-based taxonomy is

---

> ### Author Response · Authors · 2025-05-22
> **Response to Reviewer HFqA**
>
> Dear Reviewer HFqA,
>
> We sincerely appreciate your positive and encouraging feedback. We greatly appreciate your recognition of the clarity and readability of our survey, as well as the intuitive nature of our taxonomy based on stochasticity. In response to the only concern you raised, we would like to offer a detailed explanation:
>
> While prior surveys in the FFS domain have categorized models based on architectural paradigms (e.g., RNNs, CNNs, Transformers) [1], to our knowledge, no existing survey has systematically classified FFS approaches from the perspective of how they model stochasticity—i.e., their ability to handle uncertainty in future outcomes.
>
> We believe this perspective is both timely and meaningful. In real-world video forecasting and generation, uncertainty is inherent and unavoidable. Models differ substantially in their capacity to represent such uncertainty—from deterministic models that output a single likely future, to stochastic models that incorporate randomness via latent variables or sampling, to fully generative models trained to learn a distribution over plausible futures. By structuring our taxonomy around this axis, we aim to provide a deeper understanding of what different methods assume about the future and how this impacts their practical usage.
>
> We have implemented several revisions and clarifications throughout the paper. We believe this survey can benefit the community and support future research in this rapidly evolving area. We would be grateful for any further suggestions you might have.
>
>
> Ref:
> [1] Oprea, Sergiu, et al. "A review on deep learning techniques for video prediction." IEEE Transactions on Pattern Analysis and Machine Intelligence 44.6 (2020): 2806-2826.

---

> > ### Comment · Reviewer_HFqA · 2025-06-06
> >
> > Thank you for your responses. I recommend accepting the paper.

---

### Review · Reviewer_6qW7 · 2025-05-07

**Summary Of Contributions:**

This paper provides a survey on future frame synthesis by introducing the dataset, methods and applications related to FFS tasks. The survey divide the methods of the topic into three aspects, deterministic, stochastic and generative models. The survey focuses on exploring different methods for FFS.

**Audience:**

Yes

**Broader Impact Concerns:**

I do think it is worth mentioning the impact concerns since content creation/generation can easily cause/raise safety/fairness issues.

**Claims And Evidence:**

No

**Requested Changes:**

I have listed the requested changes in the weakness section. Please reference it.

**Strengths And Weaknesses:**

Strengths:
Overall, the survey is valuable to the community. It provides extensive explanations on the latest future frame synthesis methods since this topic has been one of the most attractive and critical venue in content creation and generation field.

The authors have conducted extensive surveys on methods, from deterministic, stochastic to generative models, to illustrate current mainstream models.

Weakness:

1. In section 2.2, since authors mentioned broad methods in three categories, I think it would be great to briefly mention all types of models, transformers, diffusions, flow matching, etc.

2. in 2.3.2, it is vague to simply mention short-term or long-term video synthesis. Since with the capabilities of the diffusion models, the video length can be extremely extended and major large foundational video models have shown impressive results in predicting/generating long videos. It is better to clarify and provide more context, how many seconds do short-term videos have?

3. Since this survey focuses on methodology, while I think it is worth discussing beyond architecture details, for example, how multimodal information is handled? Whether cross-attention layers have been used?

4. For equation (1), (2), I think it is a bit vague. I would recommend also consider modalities (not just controls) and refine equations in a general format.

5. Overall, since authors mentioned that this survey focuses on method, but I think this survey misses too many details of method/models. For instance, it would be great to explain details of self-attention and how successful it is, similarly to convolutions.

6. Minor comment: given the facts that we have seen a number of large foundational video generation models from industries. It would be great to summarize technologies used by industries as well.

---

> ### Author Response · Authors · 2025-05-22
> **Response to Reviewer 6qW7**
>
> Dear Reviewer 6qW7,
>
> We sincerely appreciate your precious time and constructive comments. Your feedback significantly helped us improve the clarity and technical depth of our survey. The changes can be seen in the newest version of the paper, where they are highlighted in blue. In the following, we would like to answer your concerns separately.
>
> **Weakness 1**: In Section 2.2, since authors mentioned broad methods in three categories, I think it would be great to briefly mention all types of models, transformers, diffusions, flow matching, etc.
>
> **Response**: We have revised `Section 2.2` to explicitly include representative architectures across the three paradigms. Specifically, we add transformer models such as ViT, Swin Transformer, IPT, and TimeSformer. We also incorporate diffusion-based methods (e.g., DDPM, Video Diffusion, SVD) and flow-matching approaches (e.g., Lipman et al., 2022; Dao et al., 2023).
>
> **Weakness 2**: In Section 2.3.2, it is vague to simply mention short-term or long-term video synthesis. Since with the capabilities of the diffusion models, the video length can be extremely extended and major large foundational video models have shown impressive results in predicting/generating long videos. It is better to clarify and provide more context, how many seconds do short-term videos have?
>
> **Response**: We agree with this point. We have clarified in `Section 2.3.2` that short-term video prediction models like DMVFN typically operate on horizons of approximately 0.3 seconds. In contrast, diffusion models such as SVD can now generate coherent sequences of 2–4 seconds. This temporal context now better illustrates the distinction between short- and long-term synthesis.
>
> **Weakness 3**: Since this survey focuses on methodology, while I think it is worth discussing beyond architecture details, for example, how multimodal information is handled? Whether cross-attention layers have been used?
>
> **Response**: Thank you for the insightful suggestion. We have expanded our discussion in `Section 5.1` to describe how multimodal information (e.g., text instructions, depth and segmentation maps, user controls) is integrated into FFS pipelines.
>
> **Weakness 4**: For equation (1), (2), I think it is a bit vague. I would recommend also consider modalities (not just controls) and refine equations in a general format.
>
> **Response**: We have revised and expanded the formal problem formulation in `Section 2.1`. Equations (1) and (2) now include modality variables $M_{t_1: t_2}$ and control signals $C_{t_2 + 1: t_3}$. We also added Equation (3) to generalize the formulation for multi-modal learning targets. These changes provide a more comprehensive and flexible representation for modern FFS setups.
>
> **Weakness 5**: Overall, since authors mentioned that this survey focuses on method, but I think this survey misses too many details of method/models. For instance, it would be great to explain details of self-attention and how successful it is, similarly to convolutions.
>
> **Response**: We appreciate this observation and have addressed it by substantially expanding `Section 3.1.4`. We explained the principles and advantages of self-attention in Transformers (e.g., ViT), particularly in modeling long-range spatio-temporal dependencies.
>
> **Weakness 6**: Minor comment: given the facts that we have seen a number of large foundational video generation models from industries. It would be great to summarize technologies used by industries as well.
>
> **Response**: We have added a summary in `Section 5.1` and `Table 5` that outlines recent large-scale open-sourced industry models, such as HunyuanVideo (13B), Step-Video (30B), and Seaweed-7B (7B). We describe key techniques such as chunk-wise auto-regressive diffusion, latent flow modeling, and text-guided generation. These additions provide readers with a clearer view of industrial practices.
>
> **Broader Impact Concerns**: I do think it is worth mentioning the impact concerns since content creation/generation can easily cause/raise safety/fairness issues.
>
> **Response**: We fully agree. We have added a `Broader Impact` paragraph to discuss ethical concerns such as misinformation, bias propagation, copyright risks, and job displacement. We also highlight mitigation strategies, including watermarking, inappropriate content filtering, and detection of AI-generated media (e.g., Zou et al., 2025).

---

> > ### Comment · Reviewer_6qW7 · 2025-06-04
> >
> > Thanks authors for providing additional details.

---

### Decision · Action_Editor_RgBf · 2025-06-13

**Recommendation:** Accept with minor revision

**Additional Comments:**

This paper presents a comprehensive survey on future frame synthesis (FFS), tracing the field’s progression from deterministic approaches to generative methods. After a discussion of existing paradigms, emerging trends, and widely used datasets, the authors propose a taxonomy based on three major categories: deterministic synthesis, stochastic synthesis, and generative synthesis. For each paradigm, the paper includes a comprehensive discussion of existing works and highlights relevant open challenges.

The initial reviews for this paper were cautiously positive but also raised important concerns. Reviewers acknowledged both the significance of the topic and the value and clarity of the survey, but also identified notable gaps, such as the limited discussion of video generation evaluation, connections to representation learning, and other surveys. Missing (technical) details were pointed out as well, including summaries of relevant model families, network layers, and context integration techniques. The authors responded constructively to these concerns and addressed all of them in their latest revision. Following the author response, all three reviewers either recommend accepting the paper or express an inclination to do so. I share this overall sentiment and believe the paper will be of interest to the community. However, before the paper can be accepted, I would ask the authors to make the following changes, which will further improve the quality of the paper:

• The paper would benefit immensely from a figure that visually illustrates the proposed taxonomy.

• The survey would benefit from a deeper analysis of common themes among approaches within each (sub-)category — specifically, how these methods are conceptually linked, how they group together, and how they motivate or build upon one another. I strongly encourage the authors to revisit the relevant sections with this in mind.

• I appreciate the substantial revisions the authors made during the review process. While the newly added content (highlighted in blue) addresses the reviewers’ concerns, the transitions between the original and revised sections remain somewhat abrupt and would benefit from further polishing for better narrative flow.

• The discussion of other surveys in the related work section is too disconnected from this survey’s contributions. Please discuss similarities and differences with related surveys alongside their description.

Once these changes have been implemented, the paper will be ready for acceptance.

**Audience:**

Yes

**Audience Explanation:**

This survey paper is based on a taxonomy that highlights the transition from deterministic methods to stochastic/generative methods in the  field of future frame synthesis. Beyond this high-level evolution, the community representing the field will also appreciate the discussion of emerging trends and open challenges, as well as the overview of existing metrics, datasets, and open-source initiatives.

**Claims And Evidence:**

Yes

**Claims Explanation:**

Each of the three major categories surveyed in this paper is accompanied by a comprehensive table of published works, including their references, publication venues, and core ideas.

---

> ### Author Response · Authors · 2025-07-14
>
> Dear AE,
>
> We have revised the paper according to your suggestions and submitted a non-anonymous camera-ready version. Thank you again for your efforts.
>
> Best wishes,
> TMLR Paper4528 Authors